# Lagrangian simulation of ice particles and resulting dehydration in the polar winter stratosphere

Ines Tritscher[1], Jens-Uwe Grooß[1], Reinhold Spang[1], Michael C. Pitts[2], Lamont R. Poole[3], Rolf Müller[1], and Martin Riese[1]

[1]Institute of Energy and Climate Research: Stratosphere (IEK-7), Forschungszentrum Jülich, 52425 Jülich, Germany
[2]NASA Langley Research Center, Hampton, Virginia 23681, USA
[3]Science Systems and Applications, Inc., Hampton, Virginia, 23666, USA

**Correspondence:** Ines Tritscher (i.tritscher@fz-juelich.de)

**Abstract.** Polar stratospheric clouds (PSCs) and cold stratospheric aerosols drive heterogeneous chemistry and play a major role in polar ozone depletion. The Chemical Lagrangian Model of the Stratosphere (CLaMS) simulates the nucleation, growth, sedimentation, and evaporation of PSC particles along individual trajectories. Particles consisting of nitric acid trihydrate (NAT), which contain a substantial fraction of the stratospheric nitric acid ($HNO_3$), were the focus of previous modeling

5    work and are known for their potential to denitrify the polar stratosphere. Here, we carried this idea forward and introduced the formation of ice PSCs and related dehydration into the sedimentation module of CLaMS. Both processes change the simulated chemical composition of the lower stratosphere. Due to the Lagrangian transport scheme, NAT and ice particles move freely in three-dimensional space. Heterogeneous NAT and ice nucleation on foreign nuclei as well as homogeneous ice nucleation and NAT nucleation on preexisting ice particles are now implemented into CLaMS and cover major PSC formation pathways.

    We show results from the Arctic winter 2009/2010 and from the Antarctic winter 2011 to demonstrate the performance of the model over two entire PSC seasons. For both hemispheres, we present CLaMS results in comparison to measurements from the Cloud-Aerosol Lidar with Orthogonal Polarization (CALIOP), the Michelson Interferometer for Passive Atmospheric Sounding (MIPAS), and the Microwave Limb Sounder (MLS). Observations and simulations are presented on season-long and

15    vortex-wide scales as well as for single PSC events. The simulations reproduce well both the timing and the extent of PSC occurrence inside the entire vortex. Divided into specific PSC classes, CLaMS results show predominantly good agreement with CALIOP and MIPAS observations, even for specific days and single satellite orbits. CLaMS and CALIOP agree that NAT mixtures are the first type of PSC to be present in both winters. NAT PSC areal coverages over the entire season agree satisfactorily. However, cloud free areas, next to or surrounded by PSCs in the CALIOP data, are often populated with NAT

20    particles in the CLaMS simulations. Looking at the temporal and vortex averaged evolution of $HNO_3$, CLaMS shows an uptake of $HNO_3$ from the gas into the particle phase which is too large and happens too early in the simulation of the Arctic winter. In turn, the permanent redistribution of $HNO_3$ is smaller in the simulations than in the observations. The Antarctic model run shows too little denitrification at lower altitudes towards the end of the winter compared to the observations. The occurrence of synoptic-scale ice PSCs agree satisfactorily between observations and simulations for both hemispheres and the simulated

vertical redistribution of water vapor ($H_2O$) is in very good agreement with MLS observations. In summary, a conclusive agreement between CLaMS simulations and a variety of independent measurements is presented.

## 1 Introduction

The representation of PSCs in global models is often poor despite their importance for ozone chemistry in polar winter and spring. In the lower stratosphere, PSCs provide surfaces for heterogeneous reactions activating chlorine reservoir species and thus accelerating ozone loss (Solomon et al., 1986; Crutzen et al., 1992; Solomon, 1999). Even though the importance of liquid particles with respect to chlorine activation and ozone depletion has been shown in recent publications (Drdla and Müller, 2012; Wohltmann et al., 2013; Kirner et al., 2015), solid particles influence heterogeneous chemistry substantially (Solomon et al., 2015). Especially sedimentation of solid PSC particles irreversibly changes the chemical composition of the lower stratosphere and alters the process causing ozone depletion through denitrification (Fahey et al., 2001; Molleker et al., 2014) and dehydration (Kelly et al., 1989). Further, also the uptake of nitrogen containing species in PSCs changes the chemical composition of the lower stratosphere under PSC conditions substantially, with important impact for ozone loss chemistry (Solomon, 1999; Wohltmann et al., 2017; Müller et al., 2018). Finally, model results for different types of particles are not additive in a simple way as shown by Solomon et al. (2015). Therefore, particle surface areas in models should be described as precisely and realistically as possible.

Thanks to the Montreal Protocol and its amendments and adjustments, concentrations of ozone depleting substances are now decreasing continuously (WMO, 2014, 2018) and Solomon et al. (2016) now present evidence that the healing of the Antarctic ozone layer has actually started. However, recent years showed new record ozone losses above the Arctic winter pole (Manney and Lawrence, 2016), a crucial reason to still step up efforts in understanding and modeling PSC formation on global scales better. Facing climate change, an in depth understanding of atmospheric processes becomes even more important and a complete and comprehensive knowledge of processes affecting stratospheric ozone is required to reliably predict the future evolution of the stratospheric ozone layer.

PSCs are supposed to consist of liquid supercooled ternary solution (STS) droplets, solid nitric acid trihydrate (NAT) particles and/or solid ice particles (Peter and Grooß, 2012). Their formation mechanisms are still a focus of research, newly motivated by global, high resolution satellite observations (Spang et al., 2018; Pitts et al., 2018). Due to unknown processes in the formation of solid PSC particles, large differences in the parameterization of PSCs in global models exist. Further, a detailed PSC formation scheme may require large computing times and therefore is not applicable in every model. Using PSC schemes of different complexity, the representation of PSCs in models varies as well. Most current global models use a simplified PSC scheme that prescribes number densities and particle radii and assumes thermodynamical equilibrium (Morgenstern et al., 2017). Some Chemistry Climate Models (CCMs) like SD-WACCM and EMAC (ECHAM5/MESSy Atmospheric Chemistry model) offer submodels with more detailed PSC schemes. Those can be coupled to the standard model setup for intensive

PSC studies, as done by Kirner et al. (2011, 2015) with EMAC and Zhu et al. (2015, 2017b, a) with SD-WACCM. However, as presented recently by Khosrawi et al. (2018), comparisons of measured PSC properties with corresponding EMAC results show deficiencies. Before Zhu et al. (2017a), the microphysical model for ice particles within SD-WACCM was missing and the fact that NAT nucleation in SD-WACCM is still based on the homogeneous surface nucleation scheme by Tabazadeh et al. (2002) is a matter of debate (Peter and Grooß, 2012). Non satisfying agreement between models and observations as well as fundamental differences e.g. in the NAT nucleation exist even in advanced PSC schemes, which further motivated the research presented in this paper.

Here, we present new developments extending the sedimentation module of the Chemical Lagrangian Model of the Stratosphere (CLaMS). We added ice PSC particles to complete the Lagrangian PSC scheme, which allows comparisons to PSC measurements and simulations of de- and rehydration in the Arctic and Antarctic to be performed. To demonstrate the performance of the new CLaMS ice sedimentation module, we have chosen two distinct winters, one Arctic and one Antarctic winter. The Arctic winter 2009/2010 shows widespread ice PSCs during mid January and is therefore ideally suited to test our new ice scheme. Additionally, the Arctic winter has been the focus of the intensive RECONCILE aircraft campaign, which took place from January until March 2010 (von Hobe et al., 2013). The 2011 Antarctic winter is representative for other Antarctic winters (see Fig. 3.5 in WMO, 2014). Global satellite data from the Cloud-Aerosol Lidar with Orthogonal Polarization (CALIOP) (Pitts et al., 2018) and the Michelson Interferometer for Passive Atmospheric Sounding (MIPAS) (Spang et al., 2018) are available for both winters and allow a comprehensive evaluation of our model results to be conducted.

## 2 CLaMS model description and setup

The Chemical Lagrangian Model of the Stratosphere (CLaMS) is a global, three-dimensional Chemical Transport Model (CTM) based on the Lagrangian principle (McKenna et al., 2002b, a; Konopka et al., 2004). CLaMS is structured into modules, which can individually be switched on and off as needed. The principal CLaMS modules are the trajectory module, the mixing module and the chemistry module. Within the CLaMS trajectory module, air parcels are advected forward in time based on prescribed wind fields. This study makes use of wind and temperature fields from ERA-Interim analyses provided by the European Centre of Medium-Range Weather Forecasts (ECMWF) (Dee et al., 2011). Total diabatic heating rates are also taken from ERA-Interim and used to determine vertical velocities (Ploeger et al., 2010). CLaMS uses a hybrid vertical coordinate ($\zeta$). At pressure levels lower than $300\,\mathrm{hPa}$, $\zeta$ can be interpreted as potential temperature ($\theta$). Towards higher pressure levels, $\zeta$ transforms from an isentropic to a pressure-based coordinate (Pommrich et al., 2014). Mixing is induced where the underlying wind field shows large shear, diagnosed by the Lyapunov exponent (McKenna et al., 2002b). Wherever the Lyapunov exponent exceeds a critical value, mixing between air parcels is introduced either by adding or by merging air parcels in the case of divergence or convergence, respectively. Using this method, a critical Lyapunov exponent of $1.5\,\mathrm{day}^{-1}$ ensures an approximate equally distributed grid and appropriate mixing strength (McKenna et al., 2002b). The sensitivity of simulated trace gas distributions in the upper troposphere / lower stratosphere to the value of the critical Lyapunov coefficient is further discussed

in Riese et al. (2012). Stratospheric chemistry within the CLaMS chemistry module is an updated version of McKenna et al. (2002a) with additional reactions listed in Grooß et al. (2014). The chemical composition of the whole atmosphere is described by the individual air parcels in the shown setup. Each air parcel represents a certain volume of its surrounding atmosphere with about 380 000 air parcels in total. Here, simulations were carried out with a horizontal resolution of 100 km in the polar regions. Vertically, the model is divided into 32 levels between 320 and 900 K, resulting in a vertical resolution of 400 m at 10 km up to about 800 m between 12 and 24 km altitude. The chemical initialization of the hemispheric runs starting on 01 December 2009 and 01 May 2011, respectively, was based on satellite data and observed tracer correlations (Grooß et al., 2014). Additionally, we used data from a multi-annual CLaMS simulation with simplified chemistry (Pommrich et al., 2014). The details for the individual species are given in Grooß et al. (2014) for the Arctic winter 2009/2010 and in Grooß et al. (2018) for the Antarctic winter 2011.

The CLaMS cirrus module is used in the regular CLaMS setup to calculate the dehydration of air masses at the tropopause level. This mechanism can be implemented by either using a temperature dependent parametrization for heterogeneous ice freezing (Krämer et al., 2009) or by a fixed value of 100 % for saturation over ice. Water ice is removed if the inferred particle fall speed exceeds a prescribed threshold value. This parametrization also allows dehydration in the stratosphere to be simulated (e.g. Grooß et al., 2014). However, it does not allow for simulations of vertical redistribution of water, since water is irreversibly removed once a critical fall speed is exceeded. Therefore, we restricted the cirrus module to the troposphere ($\zeta < 380$ K and PV $< 2$) within this study with an ice freezing threshold of 100 %. Within the stratosphere, the new ice parameterization of the sedimentation module takes over.

## 2.1 Sedimentation module

The CLaMS sedimentation module offers the possibility to enhance simulations of the polar winter stratosphere by PSC cloud formation and corresponding particle sedimentation. The module has been developed by Grooß et al. (2005) and was so far limited to the formation of NAT particles. Within this study, we expand the sedimentation module to the simulation of ice PSC particles. This step enables the simulation of the water redistribution. Moreover it provides the opportunity for detailed comparisons of simulated PSC properties with various observations. Finally, NAT and ice particle surface areas are calculated within the CLaMS sedimentation module and now also transfered to and used within the chemistry module. First applications are shown in a paper by Grooß et al. (2018).

Sedimenting particles in CLaMS are also described by a Lagrangian approach. So-called particle parcels are initialized in addition to CLaMS air parcels and move independently within the three-dimensional space. Every particle parcel represents a number of NAT or ice particles, equally distributed over a certain volume of air. The given number density remains constant during the particle's lifetime. Growth, sedimentation, and evaporation of the particles are carried out following the procedure described in detail in Carslaw et al. (2002). Vapor pressures of $HNO_3$ are calculated following Hanson and Mauersberger (1988), $H_2O$ vapor pressures are calculated according to Murphy and Koop (2005). Uptake and release of $H_2O$ and $HNO_3$ is

carried out by taking into account a weighted distance to the three nearest neighbors each above and below (Konopka et al., 2004). To draw contour lines of frost and NAT point temperatures, we used total abundances of $H_2O$ and $HNO_3$ from CLaMS and ERA-Interim temperatures. Further details to the module's fundamentals can be found in Grooß et al. (2005).

Currently, the sedimentation module comprises the following nucleation pathways:

– Heterogeneous NAT nucleation

Grooß et al. (2005) started the sedimentation module using a constant NAT nucleation rate taken from Voigt et al. (2005). With a rate of $7.8 \times 10^{-6} \, \mathrm{cm^{-3} h^{-1}}$, NAT formed instantaneously as soon as temperatures dropped below $T_{\mathrm{NAT}}$. In Grooß et al. (2014), the heterogeneous NAT nucleation was updated motivated by results obtained in the RECON-
CILE field campaign and new scientific findings about heterogeneous PSC nucleation. According to Hoyle et al. (2013), a saturation-dependent, non constant nucleation rate of NAT particles was formulated and used to improve the simulations. The active site theory (Marcolli et al., 2007) represents the basis for this approach. The idea behind this is that particles may offer a certain probability to nucleate NAT or ice, respectively. The probability differs from particle to particle, which leads to nucleation events over a broad temperature range as observed by Marcolli et al. (2007). So called active
sites, particle surface inhomogeneities, are assumed to initiate nucleation. A particle might carry several of these sites but only the best active site is of importance and triggers the nucleation event. For the use within CLaMS, the number of particles carrying a particular contact angle is tabulated in steps of $0.1°$ and described by a combination of temperature and saturation ratio. The nucleation rate is calculated by the sum over all bins up to the actual temperature and saturation ratio. Further particle nucleation takes place only if the temperature drops and/or the saturation ratio increases.

Compared to Grooß et al. (2014), we changed two details in the calculation. (1) Information related to the number of activated contact angles are stored for each particle parcel and are now also exchanged with the surrounding air parcels. Increasing saturation ratios increases the number of activated contact angles. Mixing is considered and as soon as PSC particles evaporate, the value for activated contact angles is reset to zero. (2) Up to now, particle formation took place
every 24 hours (Grooß et al., 2014). For this reason, $S_{\mathrm{NAT/ICE}}$ and $T_{\min}$ are traced along each air parcel trajectory to make use of the daily minimum temperature and maximum saturation ratio. We kept the possibility to nucleate PSC particles only once per day, e.g. to save computing time, but also introduced the possibility to use an hourly nucleation timestep. In this case, hourly resolved values of temperature and $S_{\mathrm{NAT/ICE}}$ are taken to calculate nucleation rates of NAT and/or ice particles.

– Heterogeneous ice nucleation

Heterogeneous ice nucleation has been implemented within CLaMS in an analogous manner as heterogeneous NAT nucleation. Vapor pressures for ice are calculated following Murphy and Koop (2005). Depending on temperature and supersaturation with respect to ice ($S_{\mathrm{ice}}$), a fixed number of ice particles may nucleate. Ice particle number densities are determined with the help of a look-up table, as done in Grooß et al. (2014) for heterogeneous NAT nucleation. The

look-up table is illustrated in Fig. 1 and based on the same parameterization for heterogeneous ice nucleation as defined in Engel et al. (2013). A combination of temperature (x-axis) and supersaturation (color-coded) defines the number of foreign nuclei initiating ice particle nucleation, which is therefore equal to the number of nucleated ice particles (y-axis on the right side of the figure). With decreasing temperature and increasing supersaturation, more contact angles can be activated (y-axis on the left side of the figure) and the number density of nucleated ice particles increases. The size of the nucleated ice particles is often determined by equilibrium conditions. While nitric acid uptake by micron-sized particles needs hours, water is in equilibrium on the timescales of seconds (Meilinger et al., 1995). Water equilibrium depends on gas-phase water partial pressure and water vapor pressure of the aerosol particles. The water fraction in the ice particles itself is calculated as the difference of the total partial pressure of water and the saturation vapor pressure of water over ice, depending on pressure and temperature (Murphy and Koop, 2005).

The look-up table, as well as the parameterization from Engel et al. (2013), requires the existence of small-scale temperature fluctuations. Those have been introduced into CLaMS as described in Sec. 2.2. The use of synoptic-scale temperatures only, as provided by ERA-Interim, would require a reduction of the nucleation barrier. We performed several sensitivity runs with different starting contact angles (not shown) and concluded that values of $S_{\mathrm{ice}}$ need to be lowered by about 0.17 to compensate missing temperature fluctuations and to achieve similar results for PSC occurrence and dehydration. However, higher cooling rates as provided by smaller scale temperature fluctuations resolve individual PSCs better. The source of these small-scale temperature fluctuations in the atmosphere are often related to gravity waves as described in Sect. 2.2.

– Homogeneous ice nucleation

Homogeneous nucleation of ice crystals from supercooled aqueous solution droplets has been described by Koop et al. (2000). To calculate the freezing threshold within CLaMS, we introduced the critical supersaturation $S_{\mathrm{cr}}$ following Kärcher and Lohmann (2002):

$$S_{\mathrm{cr}} = 2.583 - \frac{T[K]}{207.83} \tag{1}$$

This approximation is based on Koop et al. (2000) and saves additional computation time. The particle radius of $0.25\,\mu\mathrm{m}$ given by Kärcher and Lohmann (2002) agrees well with the mean radius of STS droplets (Peter and Grooß, 2012). Homogeneous ice nucleation requires a supercooling of 3 to $4\,\mathrm{K}$ compared to the ice frost point (Koop et al., 2000; Daerden et al., 2007) and therefore high supersaturations at stratospheric polar winter conditions ($S_{\mathrm{cr}}$ is nearly constant at about 1.7 in the temperature range from $\sim 180$ - $190\,\mathrm{K}$ (Maturilli and Dörnbrack, 2006)). Such conditions can be found e.g. in an orographic lift above mountains that can induce high cooling rates that freeze the entire background aerosol population of $10\,\mathrm{cm}^{-3}$ particles (e.g. Carslaw et al., 1998a; Fueglistaler et al., 2003). Therefore, we assume that homogeneous nucleation within CLaMS results in $n(\mathrm{ice}) = 10\,\mathrm{cm}^{-3}$.

– NAT nucleation on preexisting ice particles

NAT nucleation on preexisting ice particles is an accepted and often confirmed pathway of NAT formation (Carslaw et al., 1998b; Biermann et al., 1998; Luo et al., 2003). Downstream of mountain waves, NAT supersaturations are high and clouds with NAT number densities of up to $1\,cm^{-3}$ have been observed (see Peter and Grooß, 2012, and references therein). In this study, we allow $50\,\%$ of the existing ice particles to serve as NAT nucleus with an upper limit of $n(NAT) = 1\,cm^{-3}$ per nucleation event. NAT particle radii and volume are determined assuming thermodynamical equilibrium. This calculation is a first and easy attempt to include NAT nucleation on preexisting ice particles in a global model and may need refinements in later studies.

## 2.2 Parametrization of temperature fluctuations

It has been shown in several studies that small-scale temperature fluctuations are ubiquitous in the atmosphere and play an important role in ice cloud formation (e.g. Hoyle et al., 2005; Kärcher et al., 2014; Podglajen et al., 2016). Even if synoptic-scale temperatures are above PSC formation thresholds, negative temperature excursions may trigger ice formation and rapid cooling rates may change cloud characteristics. Orographically induced temperature fluctuations are the focus of several Arctic (e.g. Carslaw et al., 1998b; Dörnbrack et al., 2002) and Antarctic research studies (e.g. Höpfner et al., 2006a; Alexander et al., 2011; Noel and Pitts, 2012; Orr et al., 2015) on PSCs. Alexander et al. (2013) quantified the proportion of PSCs due to orographic gravity wave forcing in both hemispheres. Observations from the Atmospheric Infrared Sounder (AIRS) were used by Hoffmann et al. (2017) to evaluate explicitly resolved temperature fluctuations due to gravity waves in high-resolution meteorological analyses, also in both hemispheres. Even though wave amplitudes are typically underestimated, Hoffmann et al. (2017) found that observed gravity wave patterns agree well with those simulated in the ECMWF operational analysis. Due to a resolution of $1° \times 1°$ of the underlying wind and temperature fields from ERA-Interim used in this study, we do not expect to explicitly catch wave patterns such as mountain wave ice events (see e.g. Engel et al., 2013). However, a persistence of gravity wave activity from background winds at subgrid-scales needs to be parameterized somehow in order to mimic PSC formation in general.

For CLaMS, we make use of parameterizations by Gary (2006) for the Northern Hemisphere (NH) and Gary (2008) for the Southern Hemisphere (SH), respectively. Gary (2006) showed that mesoscale temperature fluctuations increase with altitude in a systematic way. They are greatest over mountainous terrain and towards polar latitudes during winter. These dependencies are expressed in the following equations to calculate mesoscale fluctuation amplitudes (MFA), that are not present on synoptic scales:

$$\text{MFA (NH)} = \left(112 - 1.21\,\text{Latitude} + 2.20\,W \cdot \text{Latitude} + 29.0\,\text{Topography}\right) \cdot \frac{\text{Pressure [hPa]}^{-0.4}}{58.85} \tag{2}$$

$$\text{MFA (SH)} = \left(114 - 0.42\,\text{Latitude} + 0.84\,W \cdot \text{Latitude} + 29.0\,\text{Topography}\right) \cdot \frac{\text{Pressure [hPa]}^{-0.4}}{58.85} \tag{3}$$

Taking negative latitudes for the SH, we assumed a mean topographic parameter of 0.5. Equation (4) represents a corrected version of the original Equation (3) of Gary (2006) (Bruce Gary, private communication). Consequently, the parameter $W$ is calculated for each day of the year (DOY) and both hemispheres in the following way:

$$W = 0.5 \cdot \left[ 1 + \sin\left( 2\pi \cdot \frac{\mathrm{DOY} - 295}{365} \right) \right] \tag{4}$$

MFA is first converted from full-width to half-width amplitude and also from its original altitude unit $\mathrm{m}$ to the temperature unit K by a simple conversion assuming a dry adiabatic temperature behavior of $1\,\mathrm{K} = 100\,\mathrm{m}$. MFA is further scaled by random numbers originating from a normal distribution, with MFA being the standard deviation. The resulting temperature is added to the ERA-Interim synoptic-scale temperature and used for the calculations of particle nucleation for both, NAT and ice particles.

## 3    Comparison to measurements

To evaluate simulations from the new ice PSC scheme within the CLaMS sedimentation module, we compare our results with satellite measurements from the Cloud-Aerosol Lidar with Orthogonal Polarization (CALIOP), the Microwave Limb Sounder (MLS), and the Michelson Interferometer for Passive Atmospheric Sounding (MIPAS). Since April 2006, CALIOP flies on the CALIPSO satellite measuring high resolution backscatter profiles from which information about PSC composition

can be inferred. MLS was launched in July 2004 on the Aura spacecraft and delivers profiles of $HNO_3$ and $H_2O$. As part of the NASA/ESA A-Train constellation, both instruments closely follow each other along the same track describing a sun-synchronous polar orbit with a global coverage ranging from $82°\,\mathrm{N}$ to $82°\,\mathrm{S}$ (Stephens et al., 2002; Waters et al., 2006). MIPAS was operating on board the Envisat satellite from July 2002 to April 2012 measuring limb infrared (IR) spectra in the wavelength range from 4 to $15\,\mu\mathrm{m}$ (Fischer et al., 2008). The satellite operated also in a sun synchronous orbit and allowed

geographical coverage up to both poles due to additional poleward tilt of the primary mirror (usually $87°\,\mathrm{S}$ to $89°\,\mathrm{N}$). MIPAS measured PSCs at day and night time.

### 3.1    CALIOP

During the last decade, CALIOP was the basis of various studies on PSCs Pitts et al. (e.g. 2007, 2009, 2011, 2013). Within this study, we make use of CALIPSO Level 2 PSC Mask Product Version 2.0 (v2), which has recently been introduced in Pitts et al.

(2018). PSCs are detected as statistical outliers relative to the background stratospheric aerosol population (at $T < 200\,\mathrm{K}$) in perpendicular backscatter ($\beta_\mathrm{perp}$) or scattering ratio ($R$) at $532\,\mathrm{nm}$. The background threshold values ($R_\mathrm{threshold}$ and $\beta_\mathrm{perp,threshold}$) are defined as median plus one median absolute deviation. The thresholds are daily values that vary with potential temperature and are included in the CALIOP v2 PSC data files. A CALIOP measurement sample is defined to be a PSC if either $\beta_\mathrm{perp}$ or $R$ exceeds the background threshold plus an uncertainty ($\sigma$), i.e. $\beta_\mathrm{perp} > \beta_\mathrm{perp,threshold} + \sigma(\beta_\mathrm{perp})$ or $R > R_\mathrm{threshold} + \sigma(R)$. An out-

lier in $\beta_\mathrm{perp}$ is assumed to contain detectable non-spherical particles. The optical space for non-spherical particles is divided into two general regimes. A dynamical boundary ($R_\mathrm{NATlice}$) separates NAT mixtures from ice. $R_\mathrm{NATlice}$ is computed from estimates

of cloud-free MLS $HNO_3$ and $H_2O$ vapors to account for effects of dehydration and denitrification. A data point is classified as ice if $R > R_{NATlice}$ and $R < 50$, and is further classified as wave ice if $R > 50$. NAT mixtures are defined opposite of the dynamical boundary with values of $R < R_{NATlice}$. Any NAT mixture with $R > 2$ and $\beta_{perp} > 2 \times 10^{-5}\,km^{-1}sr^{-1}$ belongs to the sub-class named "enhanced NAT mixtures". Each data point that is not an outlier in $\beta_{perp}$, but is an outlier in $R$, is classified as STS. A visualization of the PSC classification can be found in the lower panels of Figs. 3 and 7. Please refer to Pitts et al. (2018) for more details of the v2 PSC classification.

A comparison of individual PSC clouds simulated within CLaMS and measured by CALIOP requires a conversion of model results into optical parameters. For every CALIOP data point, we compute a size distribution from CLaMS PSC particles within a radius of $50\,km$ around the point of measurement and attribute this to the observation. As in Hoyle et al. (2013) and Engel et al. (2013), we make use of Mie theory and T-Matrix calculations to compute scattering of light by STS, NAT and ice particles as a function of wavelength (e.g. Mishchenko et al., 2010). Prolate spheroids for both ice and NAT particles with aspect ratios of 0.9 (diameter-to-length ratio) and a refractive index of 1.31 for ice and 1.48 for NAT have been chosen (Engel et al., 2013). To fully adopt the procedure of PSC classification, we calculated $\sigma$ also for modeled values using the CALIOP noise equation as follows:

$$\sigma(\beta) = CNF \cdot \sqrt{\beta} \tag{5}$$

The CALIOP noise factor (CNF) combines different scaling factors into a single value for each CALIOP horizontal averaging scale (Liu et al., 2006). We have been using a CNF of 0.00102 for all our calculations, which corresponds to an horizontal average of $135\,km$ and therefore a best case for detection. The calculations have been done with the assumption that the parallel and perpendicular components of molecular backscatter are 0.99634 and 0.00366 times the total molecular backscatter, respectively. Finally, the following relationship is used for $\sigma(R)$:

$$\sigma(R) = |R| \cdot \sqrt{\frac{\sigma^2(\beta_{perp}) + \sigma^2(\beta_{parallel})}{(\beta_{perp} + \beta_{parallel})^2} + (0.03)^2} \tag{6}$$

The last term accounts for assumed 3 % relative uncertainty in molecular backscatter.

The procedure described above is essential to compare individual optical properties on a cloud by cloud basis. Comparisons between CLaMS, CALIOP, and MIPAS showing PSC areal coverage have been performed using CLaMS PSC surface areas. Information about the surface area density of ice, NAT, and STS particles per volume of air is available for every CLaMS air parcel. The quantity is derived from CLaMS particle parcels and interpolated onto neighboring CLaMS air parcels using a distance depending weight at every time step during the simulation. This step saves computing time and allows an easier post processing of the model results for comparisons, which do not require individual size distributions. As lower boundaries, in accordance with the CALIOP detection thresholds, we use $3.3\,\mu m^2 cm^{-3}$ for STS droplets (Carslaw et al., 1994), $0.25\,\mu m^2 cm^{-3}$ for NAT, and $0.5\,\mu m^2 cm^{-3}$ for ice particles. Values exceeding those thresholds are counted as PSCs and as a specific composition class, respectively.

## 3.2 MIPAS

The MIPAS PSC detection and classification approach is based on the combination of the well-known two-color ratio method for IR limb measurements (Spang and Remedios, 2003), the cloud index, and multiple 2-D brightness temperature difference probability density functions (Spang et al., 2016). The so-called Bayesian Classifier combines the information content of various correlation diagrams of color ratios and brightness temperature differences covering several atmospheric window regions. Finally, the classifier estimates the most likely probability that either one of the three PSC types (ice, NAT, or STS) dominates the spectral characteristics of MIPAS or defines mixed-type clouds with intermediate probabilities (40 - 50 %). The MIPclouds processor for detection and cloud parameter retrieval is presented in detail in Spang et al. (2012). Spang et al. (2016) introduced the methodology of the Bayesian Classifier (v1.2.8) for PSC cloud types. The classification method has been applied to the complete MIPAS data set (Spang et al., 2018).

Within this paper, we compared horizontal distributions of PSC composition classes for both hemispheres on single days at constant levels of potential temperature. MIPAS data have been processed as described above. For CLaMS, we calculated trajectories from the model results to map those onto the MIPAS measurement locations. As done for the comparison of PSC areal coverage (see Sect. 3.1), we used information about the surface area density of ice, NAT, and STS particles from CLaMS. The same detection thresholds ($3.3\,\mu m^2 cm^{-3}$ for STS droplets, $0.25\,\mu m^2 cm^{-3}$ for NAT, and $0.5\,\mu m^2 cm^{-3}$ for ice particles) have been applied to classify CLaMS model results for the comparison with MIPAS. Please note, that vertical sampling differences exist between MIPAS (1.5 - 3 km) and CLaMS ($\sim 700\,m$) and that measured and sampled air volumes do not perfectly match.

## 3.3 MLS

MLS provides atmospheric profiles of temperature and composition (including $H_2O$ and $HNO_3$) via passive measurement of microwave thermal emission from the limb of the Earth's atmosphere (Waters et al., 2006). Those measurements are done almost simultaneously to measurements by CALIOP since the Aura satellite flies together with CALIOP in the A-Train satellite constellation. We use version 4.2 of MLS measurements. Information about vertical and horizontal along-track resolutions as well as precision and accuracy of the data can be found in Livesey et al. (2017). In short, MLS version 4.2 measurements have typical single-profile precisions (accuracies) of 4 - 15 % (4 - 7 %) for $H_2O$ (Read et al., 2007; Lambert et al., 2007) and 0.6 ppbv (1 - 2 ppbv) for $HNO_3$ (Santee et al., 2007). Vertical and horizontal along-track resolutions are 3.1 - 3.5 km and 180 - 290 km for $H_2O$, and 3.5 - 5.5 and 400 - 550 km for $HNO_3$. For comparisons with CLaMS, we interpolated MLS parameters onto $\zeta$ levels and calculated daily averages for certain levels and equivalent latitude bins with areas of equal size.

The MLS averaging kernel has also been taken into account for additional comparisons of CLaMS model data to satellite observations. As for the comparison with MIPAS, we calculated trajectories to transfer the model results to the measurement locations. Afterwards, we applied a weighting defined by the satellite's averaging kernels using pressure as the vertical coordi-

nate and the logarithm of $H_2O$ and $HNO_3$ mixing ratios (Ploeger et al., 2013). Further details about the MLS averaging kernels are discussed in Livesey et al. (2017).

## 4 Results

Within this section, we present CLaMS results for the 2009/2010 Arctic winter as well as for the 2011 Antarctic winter in comparison to satellite observations (Sect. 3). We show plots of daily averaged PSC areal coverage over the entire winter. We compare simulations of single clouds to satellite observations and simulated $H_2O$ and $HNO_3$ concentrations to MLS measurements.

The 2009/2010 Arctic winter was on average relatively warm, however, an exceptional cold period from mid December until end of January led to the formation of widespread PSCs. At that time, the RECONCILE field campaign took place (von Hobe et al., 2013) and intensive PSC studies followed from this campaign. NAT and ice parameterizations by Hoyle et al. (2013) and Engel et al. (2013), respectively, are based on PSC observations during this particular winter. To demonstrate that both parameterizations are working in CLaMS, as well as in the original Lagrangian Zurich Optical and Microphysical box Model (ZOMM), we selected 18 January 2010 for a single cloud comparison because this day was analyzed in detail by Engel et al. (2013). Moreover, this day is in the middle of the one week period of intensive ice cloud coverage in the NH vortex with largest areas covered by ice PSCs (Pitts et al., 2011).

The 2011 Antarctic winter is amongst the colder Antarctic winters with a pronounced ozone hole (Klekociuk et al., 2014). Observations from MIPAS, CALIOP, and MLS are available throughout this entire winter. This is the first time that CLaMS simulations of dehydration and denitrification are shown in detail for a SH winter. For this reason, the Antarctic winter 2011 is presented with a series of cloud comparisons throughout the whole PSC season.

### 4.1 2009/2010 Arctic winter: comparison with observations

We start the presentation of CLaMS results with a season long and vortex wide comparison of PSC areal coverage. Daily, height resolved values of PSC areal coverage in the 2009/2010 Arctic winter are shown in Fig. 2. Between $55°$ and $90°$, we defined eight latitude bands of equal area with widths that vary in latitude from $2.3°$ (250 km) up to $12.2°$ (1340 km). The occurrence frequencies (number of PSC detections divided by the total number of observations) has been calculated for each band and altitude grid box. The PSC areal coverage is estimated as the sum of the occurrence frequency multiplied by the total area. This approach has also been used by Spang et al. (2018) and Pitts et al. (2018) and bypasses the caveat of the irregular sampling density due to the orbit geometry. MIPAS observations and CLaMS simulations are only considered at latitudes $< 82° \, N/S$ to mimic the latitudinal sampling coverage of CALIOP. The model performance at PSC altitudes for the overall winter is remarkably good (Fig. 2). PSC occurrence starts in mid of December and lasts until the end of January in both independent measurements as well as in the simulation. CLaMS shows two maxima in PSC areal coverage similar to CALIOP. Also the

vertical extent agrees well between all three panels (CALIOP, MIPAS, and CLaMS). Looking in particular at ice PSCs, CLaMS misses ice clouds in the first two weeks of January that are seen by both CALIOP and MIPAS. Those ice clouds are mountain wave induced events as shown by Pitts et al. (2011), caused by wave-driven temperature minima which are missing in the ERA-Interim data (see Fig. 8 in Engel et al., 2013). Starting in mid-January, as synoptic-scale temperatures fell below the frost

point, large areas of ice PSCs also develop within CLaMS with an extension comparable to the observations.

The disagreement between CALIOP, MIPAS, and CLaMS at altitudes below 15 km is noticeable. CALIOP observes cirrus clouds throughout the entire 2009/2010 season at altitudes below 15 km (Pitts et al., 2018). Further, CALIOP NAT at low altitudes is likely cirrus that has been misclassified. Volcanic aerosol from the Sarychev (48.1° N, 153.2° E) eruption in June 2009 were transported into the polar region producing an enhancement in the background aerosol at altitudes below about

18 km. MIPAS is highly sensitive to the presence of this volcanic aerosol which biases the MIPAS PSC detection (Spang et al., 2018) causing a striking maximum of PSC areal coverage at low altitudes in the early winter period. The widespread volcanic aerosol does produce an enhancement in the CALIOP estimate of the background levels, but will not significantly affect the PSC product since it is based on outlier detection. Both, cirrus and volcanic aerosols introduce a bias in the MIPAS PSC detection and are misclassified as NAT (Spang et al., 2018). In CLaMS, we do not simulate clouds other than PSCs. The

origin of the larger PSC area at low altitudes seen in the CLaMS PSC area panel can be explained by the altitude independent fixed detection threshold of $3.3 \, \mu m^2 cm^{-3}$ for STS droplets. At altitudes around 12 km, the stratospheric aerosol layer becomes visible as well. To reduce the large "PSC area" in CLaMS at low altitudes, we introduced a temperature threshold to this plot. Only data points with temperatures less than 200 K are considered. This temperature threshold reduces the maximum values of PSC areal coverage slightly.

A detailed look on individual clouds illustrates the performance of the CLaMS ice PSC scheme. In the middle of the one week period of synoptic-scale ice PSCs in January 2010, we selected the 18 January 2010 for a single cloud comparison (Fig. 3). This particular orbit has already been the focus of Engel et al. (2013) to adjust the heterogeneous nucleation rates for ice. Whereas Engel et al. (2013) used the microphysical box model ZOMM, run on single trajectories and starting at most ten

25    days before the point of observation, we are using a CTM in this study, initialized on 01 December 2009 and running for the whole winter 2009/2010. Despite those fundamental model differences, the result is convincing and the agreement between both models suggests a robust parameterization of heterogeneous ice nucleation. The cloud classification, as well as the individual parameters of aerosol backscatter ratio $(R-1)$ and $\beta_{perp}$, agree well with the CALIOP observations. Areas classified as ice, lie predominantly within the $T_{frost}$ contour line and areas classified as NAT within the $T_{NAT}$ contour line. The maximum values

30    of $R-1$ and $\beta_{perp}$ are in the same order of magnitude and the vertical and horizontal extent of the observed PSC agrees with the model result. The lower panels of Fig. 3 illustrate the CALIOP v2 PSC classification scheme. The dashed lines are dynamical thresholds as mentioned in Sect. 3.1. Plotted are daily maximum values for $\beta_{perp,threshold}$ and $R_{threshold}$ and the daily mean for $R_{NATlice}$. The spread in the CALIOP data is caused by measurement noise. Although measurement noise is mimicked and added to the modeled data, the spread in the modeled data is slightly less than for the observed values. Those data points are still more

35    confined and do not fill the whole space of the diagram. Enhanced NAT mixtures represent PSCs heterogeneously nucleated

in wave ice PSCs. The CALIOP criteria defining enhanced NAT mixtures are conservative and therefore, the enhanced NAT mixtures subclass is not all-inclusive (Pitts et al., 2018). On this particular day, we expect no NAT PSCs downstream of wave ice clouds. Whereas this area is not populated in the CLaMS data, single scattered measurement points from CALIOP fall into this class, likely due to measurement noise.

A comparison to MIPAS is also beneficial to validate the new CLaMS PSC scheme. A different measurement technique as well as the fact that MIPAS data has not been used to adjust the nucleation rate allows an independent quality check. Figure 4 presents the daily PSC distribution of the Bayesian classifier for 18 January 2010. The colored symbols represent the PSC classes, where in addition to the three main classes (ice, STS, NAT), a mixed type of STS and NAT (NAT_STS) particles is

10 shown. As described already in Sect. 3.2, CLaMS surface area densities have been mapped onto the MIPAS profile locations. The classification of CLaMS results is based on surface area densities, which need to be large enough to exceed an empirical detection threshold (STS: $3.3\,\mu m^2 cm^{-3}$, NAT: $0.25\,\mu m^2 cm^{-3}$, ice: $0.5\,\mu m^2 cm^{-3}$). The mixed type is chosen such that the simulated surface areas for both, STS and NAT, need to lie above the threshold. Even though ice formation is highly temperature depended, the spatial pattern of ice PSC occurrence between MIPAS and CLaMS agrees well (Fig. 4). The model

simulates ice PSCs over Spitsbergen and in the center of the cold pool above the Russian Arctic. These are the same locations where MIPAS observed ice. NAT observations downstream of ice PSCs are also reproduced by the model. However, CLaMS produces more NAT PSCs over the North Pole than observed by MIPAS. A possible explanation for this discrepancy lies in the detection threshold of MIPAS. MIPAS may classify a volume of air as STS even though NAT particles are present with volume densities smaller than about $0.3\,\mu m^3 cm^{-3}$ or NAT particles with radii larger than $3\,\mu m$ (Höpfner et al., 2006a, b; Spang et al.,

2018). Moreover, NAT clouds are mixtures of NAT and STS or even NAT, STS, and ice particles as emphasized by e.g. Peter and Grooß (2012) and Pitts et al. (2013, 2018) leading to the conclusion that the discrepancy between NAT and STS as seen in Fig. 4 might be a caveat in the MIPAS classification. However, as seen in the results for the Antarctic winter 2011, CLaMS also tends to overestimate NAT occurrences (compare Sect. 4.2).

Figure 5 demonstrates the effect of PSCs on the distribution of gas-phase $HNO_3$ and $H_2O$ of the polar winter stratosphere. First, PSCs lead to a temporary removal of $HNO_3$ and $H_2O$ from the gas phase by condensation onto NAT and ice particles, respectively, and uptake by STS droplets. This temperature removal from the gas phase can nonetheless be very important for ozone destroying chlorine chemistry. In case of sedimentation, a permanent redistribution of the gas phase components takes place. The temporal evolution of Arctic $H_2O$ is presented in Fig. 5 as function of potential temperature and time for the

average vortex core (equivalent latitude $> 70°\,N$). It shows predominantly the dynamically forced diabatic descent of air inside the polar vortex (compare Fig. 5, upper left panels). Areas of low $H_2O$ mixing ratios due to ice PSCs are too limited in the Arctic to be clearly seen in a vortex average. A tiny sign of water uptake by PSCs is visible mid of January at approximately $500\,K$ in the MLS data. This effect is even smaller in the model simulation. In contrast, the pattern of $HNO_3$ shows a clear layer of denitrified air in both observations and simulations at the end of December 2009 (compare Fig. 5, lower panels).

Additionally, a layer of renitrified air forms between approximately 400 and 450 K. The redistribution of $NO_y$ in the Arctic has already been presented but not compared to MLS observations by Grooß et al. (2014).

The panels on the right hand side of Fig. 5 show the differences between CLaMS and MLS. Bluish colors mean that CLaMS values of $H_2O/HNO_3$ are too small compared to MLS, meaning model dehydration/denitrification was too efficient. Reddish colors mean the opposite, too little dehydration/denitrification, therefore CLaMS values of $H_2O/HNO_3$ are too large compared to MLS. Hatched areas mean that CLaMS simulations with and without the sedimentation module have equal results. Differences to MLS within the hatched areas cannot be improved by changes in the ice or NAT particle nucleation and sedimentation scheme and have reasons beyond the scope of this study. The assumption that CLaMS tends to overestimate NAT occurrences is supported by the deviations in $HNO_3$ between the model and MLS. CLaMS shows an uptake of $HNO_3$ from the gas into the particle phase which is too large and happens too early in the season. Mid of January, this tendency turns around. The permanent redistribution of $HNO_3$ is smaller compared to the observations.

## 4.2 2011 Antarctic winter: comparison with observations

Figures 6 - 10 show the results for the Antarctic winter 2011 corresponding to the figures above shown for the Arctic winter 2009/2010. In addition, we present a number of days throughout the winter to demonstrate the evolution of PSCs and corresponding model simulations (Figs. 7, 8, and 9). Figure 6 shows the daily, height resolved values of PSC areal coverage and gives an overview of the PSC season in the SH 2011. Starting in the second half of May, CLaMS and CALIOP agree well with both showing NAT PSCs to be the first type of PSCs present in the season. In contrast, the MIPAS classifier detects STS clouds first, with NAT PSCs following a few days later (see also Fig. 9). Unfortunately, there are gaps in the MIPAS data record at the beginning of the PSC season. Going further in time, a large fraction of the vortex is covered by ice PSCs, which is typical for Antarctic winters. Also the agreement between CALIOP, MIPAS, and CLaMS is satisfying at PSC altitudes (Fig. 6). At altitudes below 15 km, large areas of NAT can again be seen in the MIPAS data. Those values can be attributed to cirrus clouds, which are misclassified as NAT due to a bias in the PSC detection (Spang et al., 2018). CALIOP indicates large areas of ice and some NAT mixtures below 15 km, with the NAT mixtures at these lower altitudes likely being misclassified as ice.

Throughout the PSC season, we also looked at single clouds on individual days in detail (see Fig. 7, Fig. 8, and Fig. 9). As already highlighted in the paragraph above, NAT clouds are mixtures of NAT and STS particles and MIPAS might misclassify NAT particles with radii larger than 3 μm. This discrepancy is visible in Fig. 6 and can be seen even more clearly by comparing a single day right in the beginning of the PSC season. The upper panel of Fig. 8 shows a predominantly NAT cloud seen by CALIOP on 25 May whereas Fig. 9 shows STS droplets classified by MIPAS for the entire day. Size distributions from CLaMS simulations on 25 May 2011 point in the same direction, namely that simulated NAT number densities do not exceed $10^{-2}\,cm^{-3}$ with particle radii larger than 5 μm (not shown). Discrepancies between the model and both satellite datasets appear to be related to the relative proportion of STS and NAT PSCs with CLaMS overestimating NAT occurrences. Fig. 7 shows one such example where CLaMS produces widespread NAT particles in a region where CALIOP observes few or no NAT particles. Moving the focus back to the simulation of ice PSCs, the agreement on the 25 June 2011 is convincing. The simulation of ice

is confined to areas with temperatures below $T_{\text{frost}}$. The spread in the modeled data shown in the lower panels ($1/R_{532}$ vs. $\beta_{\text{perp}}$) is slightly less than for the observed values. However, observations and model results lie close together. Overall, over the entire season, CLaMS simulations somewhat underestimate ice occurrences on several occasions (e.g. Fig. 8, July and August). However, Fig. 6 gives the impression that the areal coverage of ice PSCs is at least as large as in the observations. In general, the seasonal evolution with variations in dominating PSC types, vertical PSC occurrences and spatial patterns is reproduced by the simulations.

The temporal evolution of gas-phase water vapor and nitric acid as measured by MLS and simulated by CLaMS is presented in Fig. 10. Dehydrated and denitrified areas are clearly seen in the MLS measurements and in the CLaMS simulations. Water vapor mixing ratios as low as 1.6 ppm (vortex core average) are observed. $HNO_3$ mixing ratios in the vortex core are extremely low and reach values of 200 ppt. Evaporation of sedimenting PSC particles produces layers of enriched $H_2O$ (rehydrated) and $HNO_3$ (renitrified) air. They appear below the depleted regions as seen in both MLS observations and CLaMS. The results highlight the new capability of CLaMS to simulate the vertical redistribution of $H_2O$ in good agreement with observations. The cirrus module of CLaMS would irreversibly remove the water in the dehydrated areas. In contrast, the new sedimentation module conserves $H_2O$. The signal of sedimentation with subsequent rehydration below is visible until the end of July in the observations as well as in the simulations. Thereafter, a diabatic descent of the rehydrated layer causes water that had originally sedimented from higher latitudes to accumulate in the tropopause region. Vertical velocities are deduced from ERA-Interim diabatic heating rates. In October and November 2011, ERA-Interim already indicates positive ascent rates, causing the behavior seen in the simulations in spring. The temporal evolution of both species agrees reasonably well between MLS and CLaMS. The difference between measurements and simulations are quantified in the right panels (Fig. 10). The minimum values of $H_2O$ match very well. The layer of rehydrated air around 350 K potential temperature is slightly less than in the observations meaning that $H_2O$ mixing ratios are smaller in the simulation than in the observations. A comparison between MLS and CLaMS $HNO_3$ mixing ratios is acceptable but reveals differences. CLaMS $HNO_3$ gas phase mixing ratios around 500 K potential temperature are lower than the observations for the whole season. However, from August on, a layer of high $HNO_3$ values below 500 K points to the possibility that the redistribution of $HNO_3$ is not efficient enough in the simulation and needs to extend down to lower altitudes. This might explain the simulation of NAT particles in areas which are almost cloud free in the observations as seen in Fig. 7. Even though CLaMS gas phase mixing ratios of $HNO_3$ might be even lower than observed at that time, $HNO_3$ in the model could still be present in the particle phase and could not be redistributed correctly to lower altitudes.

## 5 Conclusions

We present CLaMS simulations based on a new Lagrangian ice sedimentation scheme focusing on the 2009/2010 Arctic and 2011 Antarctic winters. Previous CLaMS studies solely simulated NAT PSCs and the corresponding denitrification. Here, we extended the model by adding ice PSC particle nucleation, growth, sedimentation, and evaporation. Heterogeneous and ho-

mogeneous ice nucleation is included, as well as NAT nucleation on preexisting ice particles. Heterogeneous ice nucleation rates are based on Engel et al. (2013), homogeneous ice nucleation occurs if the ice saturation ratio exceeds a temperature-dependent, critical saturation (Kärcher and Lohmann, 2002). In addition, the implementation of ice particle nucleation requires small-scale temperature fluctuations (Gary, 2006, 2008) to be added to the synoptic-scale temperature, here from the ERA-Interim reanalysis.

The agreement between CLaMS simulations and the CALIOP, MIPAS, and MLS observations on different temporal and spatial scales is convincing. CLaMS PSC areal coverage presented for both seasons is in good agreement with MIPAS and CALIOP. Similar comparisons between satellite observations and model simulations have been performed in the past (e.g. Khosrawi et al., 2017, 2018; Zhu et al., 2017b, a). However, this is to our knowledge the first study presenting detailed results of individual PSCs simulated by a global model in comparison to high resolution satellite observations. In general, CLaMS tends to overestimate the occurrence of NAT PSCs. In comparison to MIPAS, CLaMS simulates NAT particles at locations where MIPAS observes liquid PSC particles. At the onset of Antarctic PSC occurrence in May, CLaMS is in agreement with CALIOP observations indicating that NAT clouds were first observed. MIPAS first observes STS, but is not sensitive to the presence of larger NAT particles (Höpfner et al., 2006a, b; Spang et al., 2018). However, the comparisons with CALIOP also shows differences regarding NAT occurrence. Cloud free areas, next to or surrounded by PSCs in the CALIOP data, are often populated with NAT particles in the CLaMS simulations. Looking at the temporal and vortex averaged evolution of $HNO_3$, CLaMS shows an uptake of $HNO_3$ from the gas into the particle phase which is somewhat too large and happens too early in the NH season. The permanent redistribution of $HNO_3$ in the NH is smaller compared to the observations. Also the Antarctic model run shows too little denitrification at lower altitudes towards the end of the winter compared to the observations. These findings point to shortcomings in the simulation of NAT particle sizes in combination with number densities namely that NAT particle sedimentation should be more efficient in CLaMS. Further studies should try to find better combinations of NAT number densities and sizes with the potential to denitrify the stratosphere more precisely. Heterogeneous NAT nucleation on foreign nuclei and preexisting ice particles is already implemented and covers most currently discussed routes to form NAT. However, NAT clouds downstream of mountain waves may act as "mother clouds" and individual NAT particles falling out of these clouds in low number densities can grow to large sizes of up to $10\,\mu m$ (Fueglistaler et al., 2002). So far, CLaMS comprises the development of high number density NAT clouds on ice surfaces. No attention has been paid to the "mother cloud" theory, which could be a step forward to resolve deviations seen in the NAT simulations.

Simulated ice PSC coverages, in turn, are almost in agreement with the observations as seen in the comparison between CALIOP, MIPAS, and CLaMS. CLaMS simulations miss wave ice clouds as seen in the beginning of the 2009/2010 Arctic winter, likely because ERA-Interim temperature fields are too coarse to resolve the required low values. The small-scale fluctuations added in this study are ubiquitous and not related to specific gravity wave sources such as mountains. A great leap forward would be a change from ERA-Interim to the higher resolution ERA5 data. This dataset may resolve mountain waves and once high supersaturations with respect to ice are reached, a revision of the homogeneous ice nucleation parameterization

is meaningful. The modeled extent of the dehydration signal fits with the MLS $H_2O$ observations. To bring this to even better agreement, the level of dehydration in CLaMS could be slightly larger, to lower the minimum values of $H_2O$ in the simulations by about $0.5\,ppm$. So far, ice nucleation on preexisting NAT particles has not been considered in our CLaMS simulations. This pathway will be included in the next CLaMS PSC study. Recently, Voigt et al. (2018) speculated about the existence of

this specific pathway, which was mentioned in the literature already two decades ago (e.g. Peter, 1997). Discussions about the importance of ice nucleation on NAT particles, e.g. for denitrification, are controversial (Khosrawi et al., 2011; Engel et al., 2014) and this topic is likely to be a focus of CLaMS studies in the future.

Despite deficiencies, the overall agreement between CLaMS and different PSC and trace gas measurements is convincing.

The advanced microphysical PSC scheme, which includes now STS, NAT, and ice PSCs, therefore represents a major improvement of the representation of cloud physics in CLaMS. Further studies will benefit from this development and may provide more insights into occurrence and importance of PSC types and their formation mechanisms.

*Code and data availability.* CLaMS code/data is available on request by the first author. Observational datasets from CALIOP, MIPAS, and MLS can be found on the following web pages (last accessed in March 2018):

MIPAS/Envisat Observations of PSCs: https://datapub.fz-juelich.de/slcs/mipas/psc/

CALIPSO/CALIOP L2 PSC Mask (v1): https://doi.org/10.5067/caliop/calipso/lid_l1-standard-v4-10

Aura MLS $H_2O$ product: https://mls.jpl.nasa.gov/products/h2o_product.php

Aura MLS $HNO_3$ product: https://mls.jpl.nasa.gov/products/hno3_product.php

*Competing interests.* The authors declare that they have no conflict of interest.

*Acknowledgements.* We thank Beiping P. Luo for sharing his Mie and T-Matrix calculations with us. We also thank Felix Ploeger and Paul Konopka for providing the data of the multi-annual CLaMS simulation. Nicole Thomas provided excellent support for our programming activities. We gratefully acknowledge the European Centre for Medium-Range Weather Forecasts (ECMWF) for providing ERA-Interim data. We thank Michelle Santee and the MLS team for providing their high quality data sets. Computing time has been granted on the

supercomputer JURECA at the Jülich Supercomputing Centre (JSC) under the VSR project ID JICG11. Ines Tritscher was funded by the Deutsche Forschungsgemeinschaft (DFG) under project number 310479827. We finally acknowledge the Stratosphere-troposphere Processes And their Role in Climate (SPARC) project and the International Space Science Institute (ISSI) for supporting the Polar Stratospheric Cloud Initiative (PSCi).

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

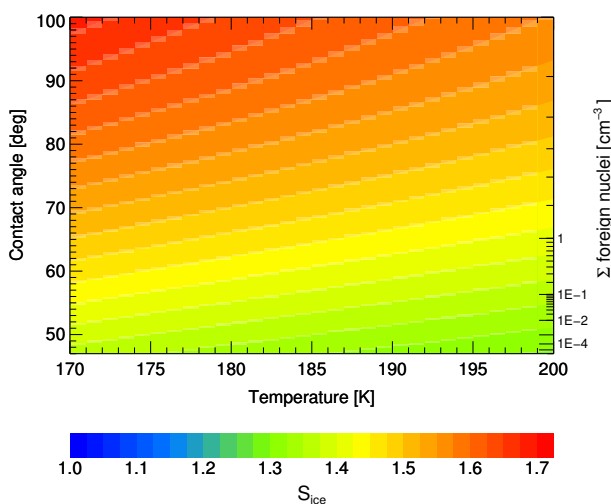

**Figure 1.** Visualization of the heterogeneous ice nucleation parameterization derived from Engel et al. (2013). The sum of foreign nuclei initiating heterogeneous ice nucleation (equal to a certain contact angle) resulting from a combination of temperature and supersaturation (color-coded).

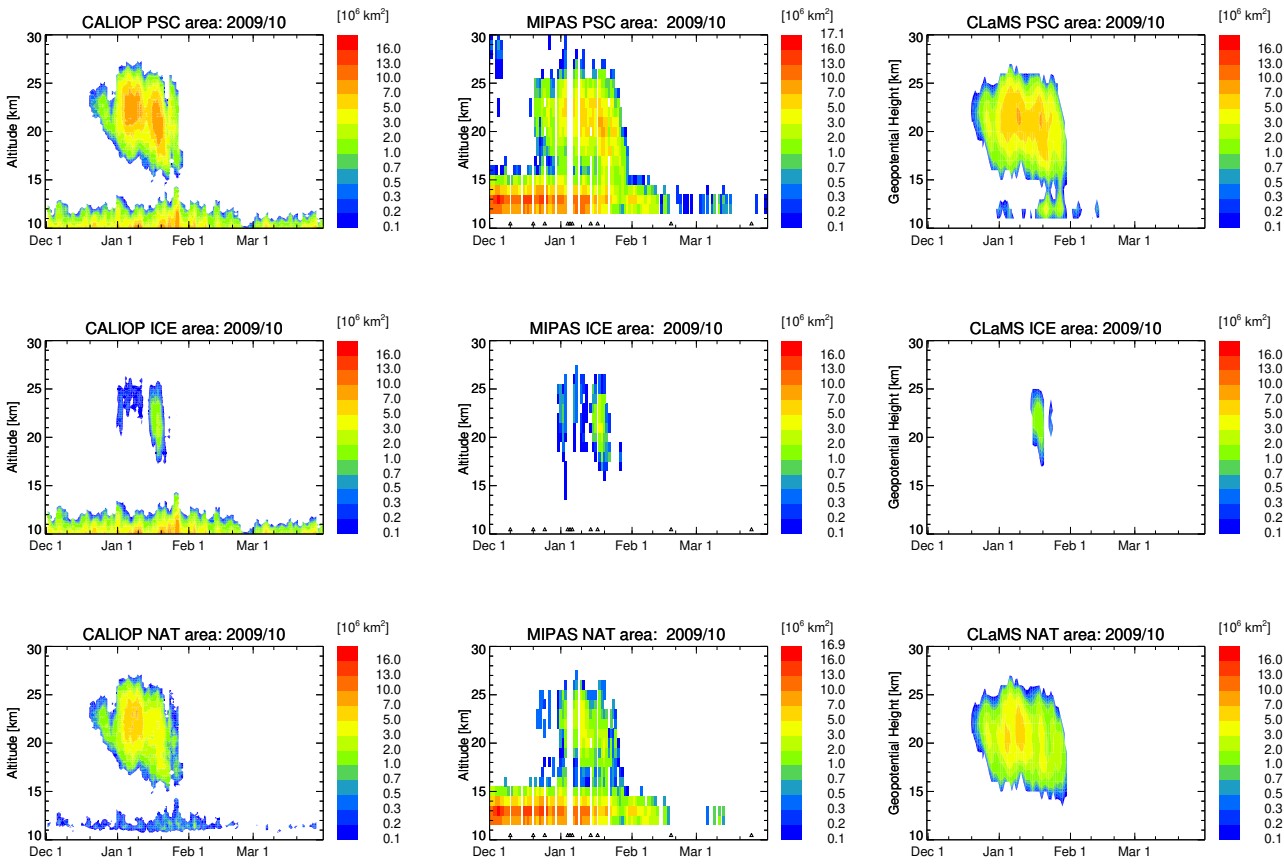

**Figure 2.** Comparison of PSC areal coverage between CALIOP (v2) left column, MIPAS (v1.2.8) middle column, and CLaMS right column from 01 December 2009 until 31 March 2010. Total PSC areal coverage ($A_{PSC}$) in $10^6 \text{km}^2$ (top row) as well as further classified ice (middle row) and NAT PSCs (bottom row) are presented as a function of time and altitude throughout the 2009/10 NH winter. MIPAS observations and CLaMS simulations are restricted to latitudes $< 82°$ N. PSC thresholds for CLaMS simulations are as follows: STS: $3.3\,\mu\text{m}^2\text{cm}^{-3}$, NAT: $0.25\,\mu\text{m}^2\text{cm}^{-3}$, ice: $0.5\,\mu\text{m}^2\text{cm}^{-3}$. Black triangles in the time series of the measurements are indicating data gaps. Please note that the color code is always identical except the maximum value of the top color bin.

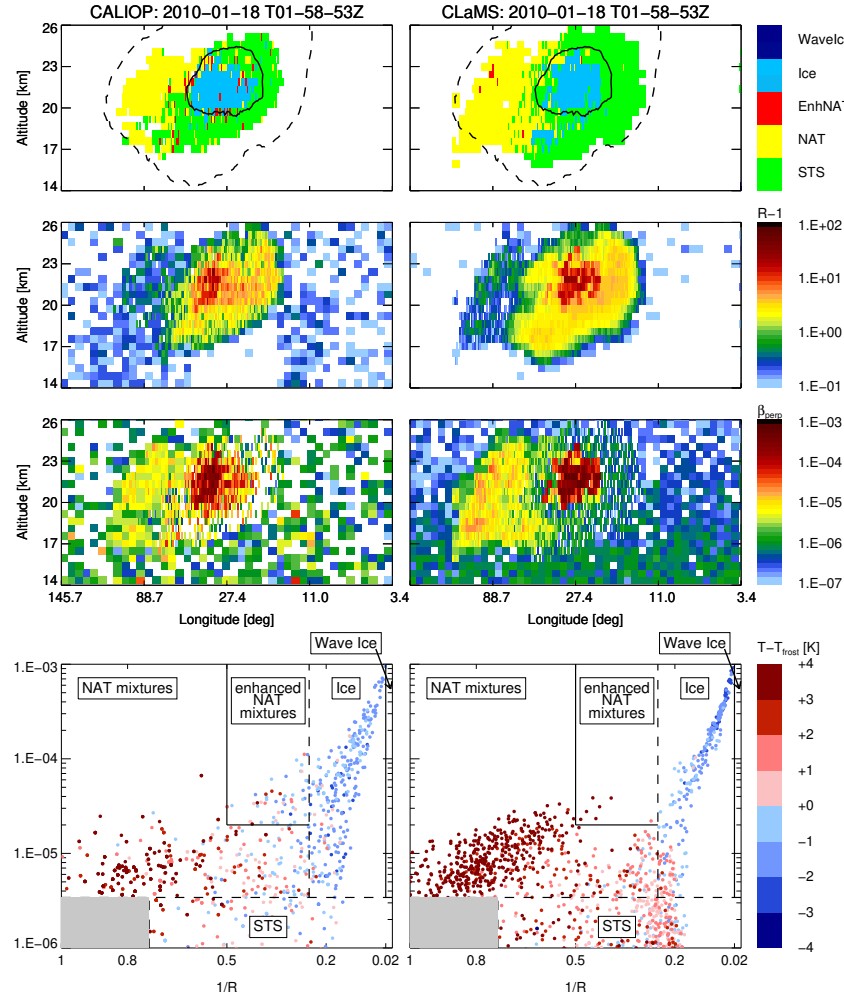

**Figure 3.** 18 January 2010: CALIPSO orbit track 2010-01-18T01-58-53Z. CALIOP measurements are shown in the left column, corresponding model results in the right column. First row: CALIOP PSC classification v2 with overlaid temperature contours for $T_{\mathrm{frost}}$ (solid line) and $T_{\mathrm{NAT}}$ (dashed line); second row: aerosol backscatter ratio ($R-1$); third row: perpendicular backscatter signal ($\beta_{\mathrm{perp}}$); lowermost row: inverse backscatter ratio ($1/R$) vs. perpendicular backscatter signal ($\beta_{\mathrm{perp}}$) with data color-coded by temperatures relative to $T_{\mathrm{frost}}$, and overlaid CALIOP v2 PSC composition classification scheme. Dashed lines are dynamical thresholds ($\beta_{\mathrm{perp,threshold}}$, $R_{\mathrm{threshold}}$, and $R_{\mathrm{NATIice}}$) (compare Fig. 4 in Pitts et al., 2018).

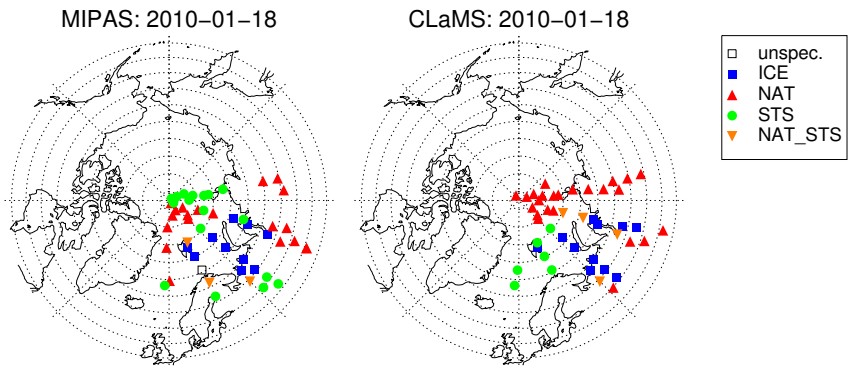

**Figure 4.** Horizontal distribution of MIPAS (left) and CLaMS (right) PSC composition classes for 18 January 2010 at an altitude level of 500 K ($\pm$ 20 K) potential temperature.

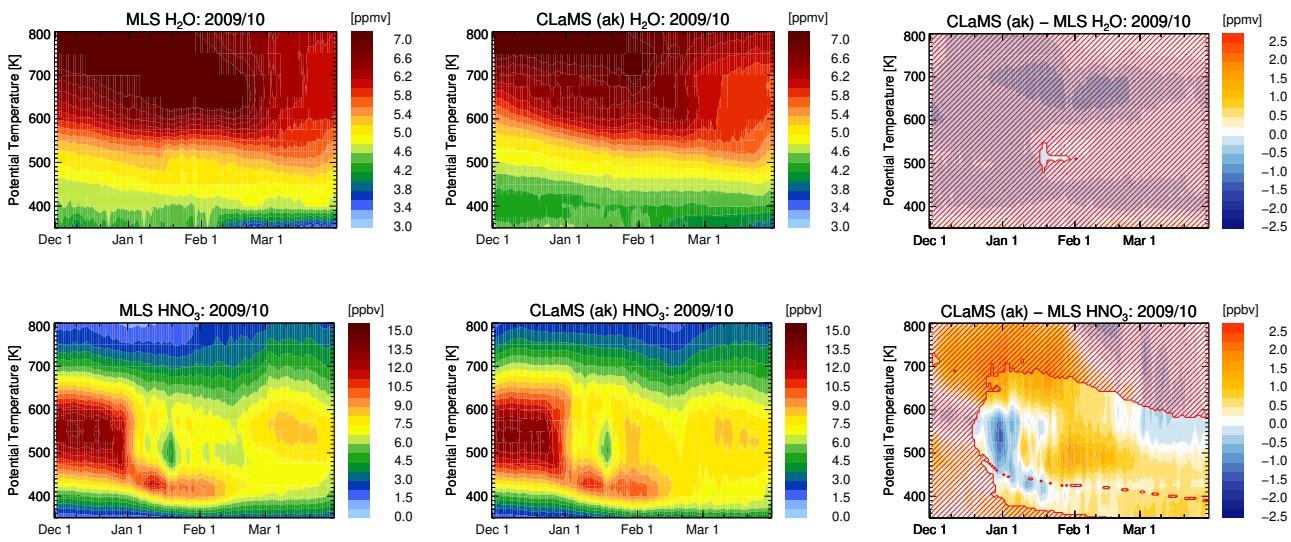

**Figure 5.** Temporal evolution of water vapor ($H_2O$, top row) and nitric acid ($HNO_3$, bottom row) are shown as an average inside the core of the polar vortex (equivalent latitudes $> 70°$ N) from 01 December 2009 until 31 March 2010. MLS measurements are presented in the left column, CLaMS model results accounting for the MLS averaging kernel in the middle column, and the difference between MLS and CLaMS in the right column. Regions not affected by any changes in the CLaMS sedimentation module are hatched.

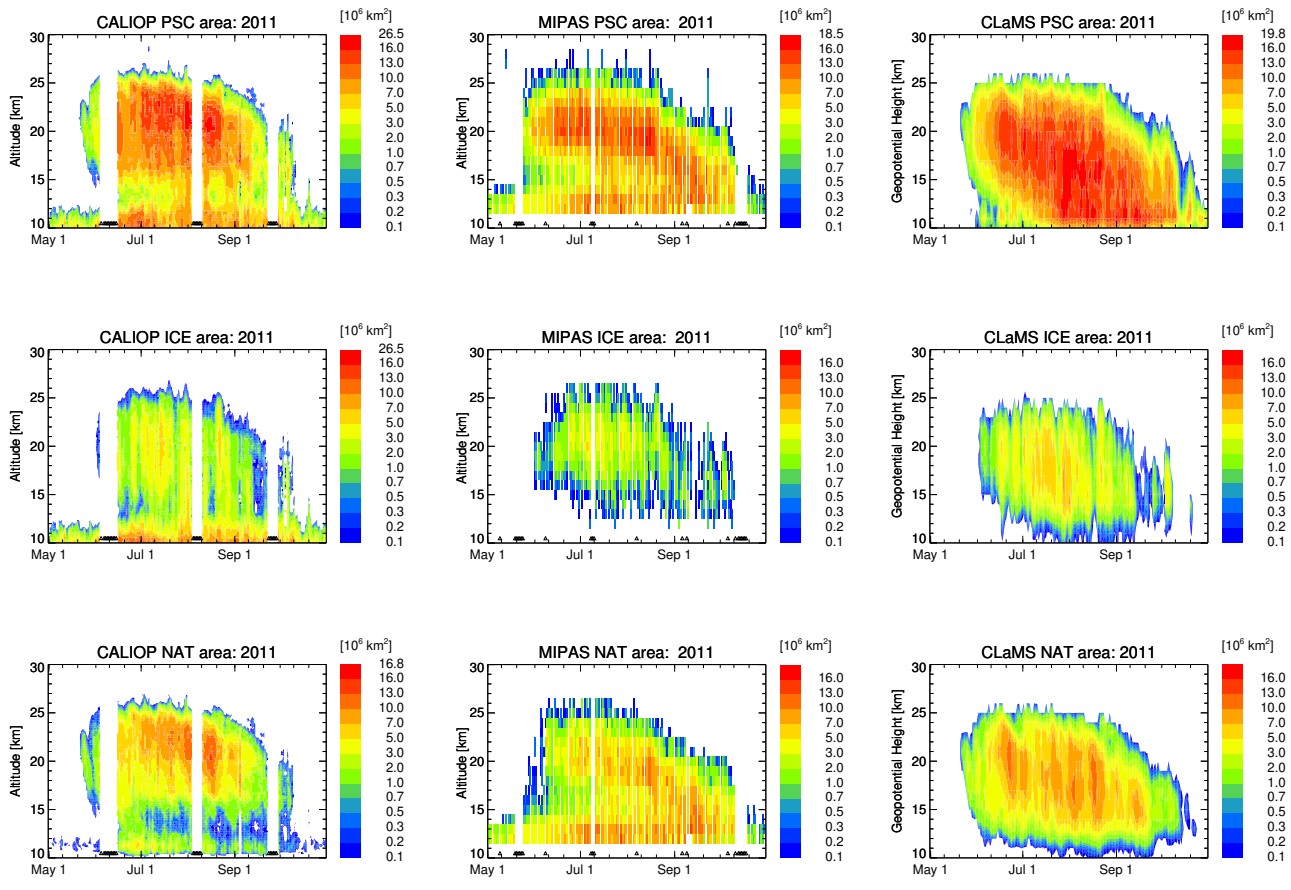

**Figure 6.** Same as Fig. 2 but for the 2011 SH winter from 01 May 2011 until 31 October 2011. Here, MIPAS observations and CLaMS simulations are restricted to latitudes $< 82°$ S. Please note that the color code is always identical except the maximum value of the top color bin.

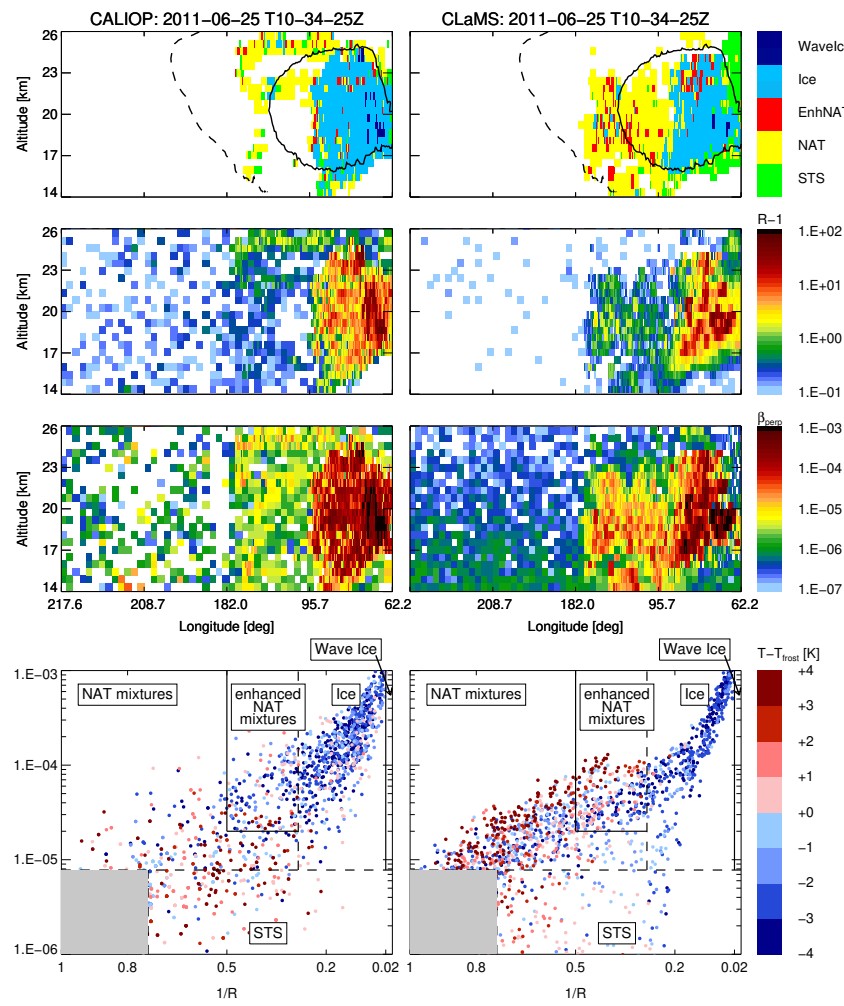

**Figure 7.** Same as Fig. 3 but for 25 June 2011: CALIPSO orbit track 2011-06-25T10-34-2Z.

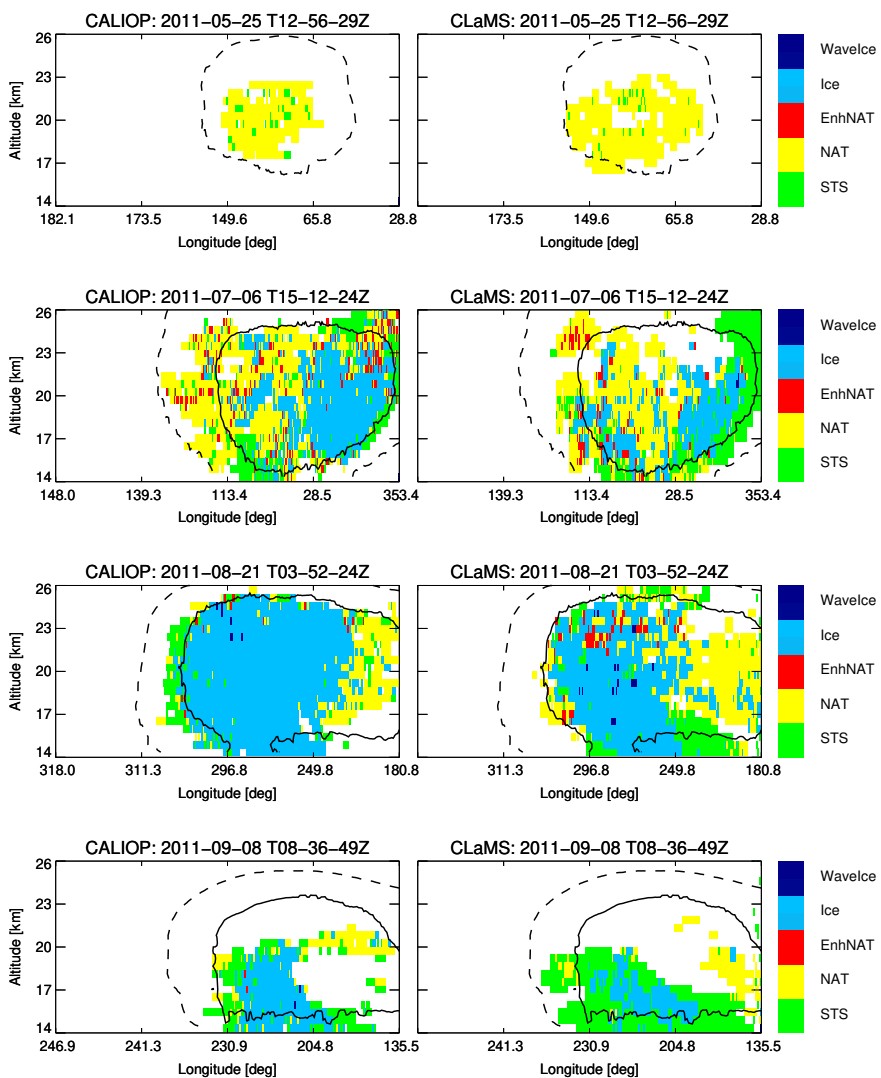

**Figure 8.** Same as first row in Figs. 3 and 7 but for four certain days in SH winter 2011: 25 May (2011-05-25 T12-56-29Z), 06 July (2011-07-06 T15-12-24Z), 21 August (2011-08-21 T03-52-24Z), 08 September (2011-09-08 T08-36-49Z).

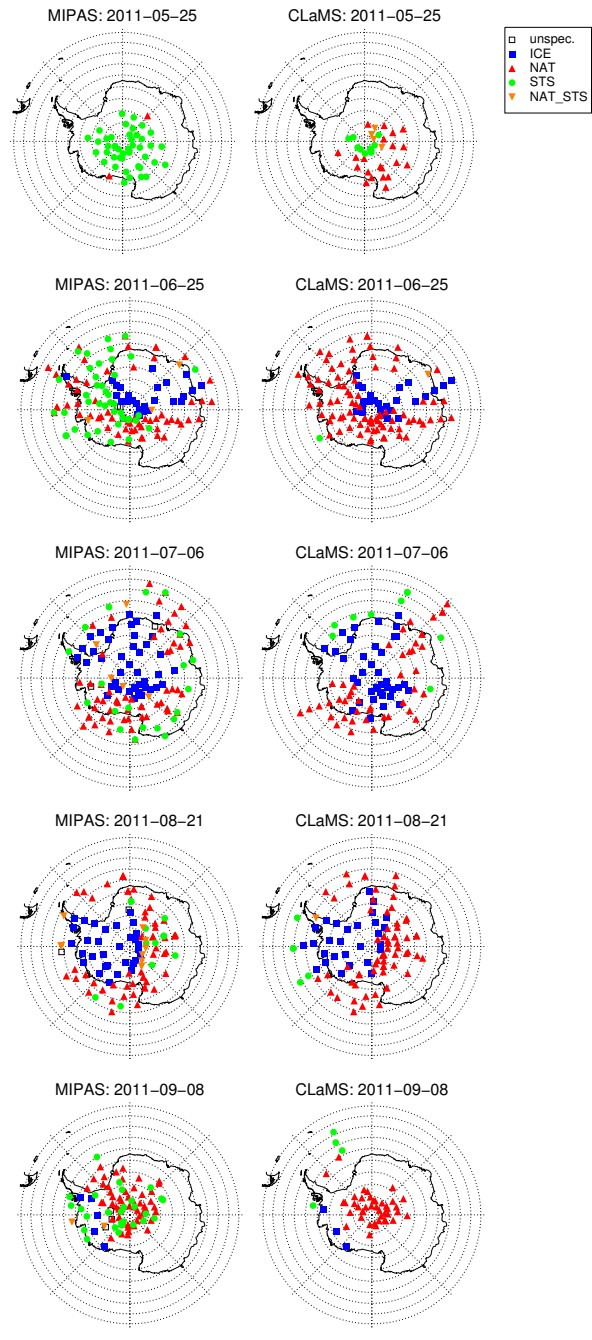

**Figure 9.** Same as Fig. 4 but for certain days in SH winter 2011: 25 May, 25 June, 06 July, 21 August, 08 September.

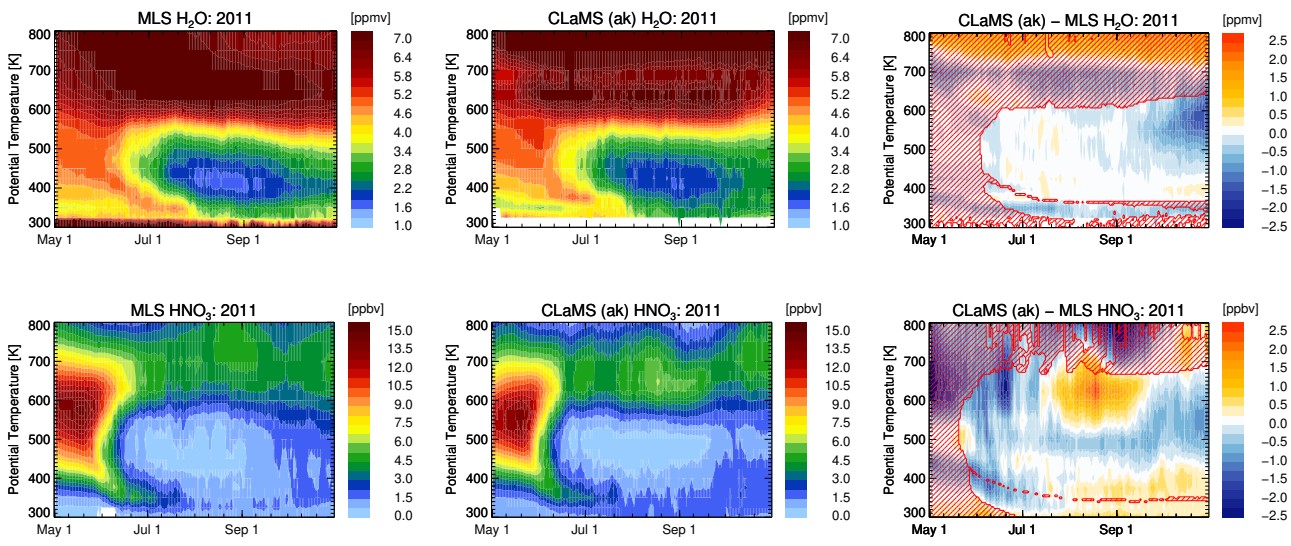

**Figure 10.** Same as Fig. 5 but for the 2011 SH winter from 01 May 2011 until 31 October 2011 and for equivalent latitudes $> 70°$ S.