# Peer review of "Lagrangian simulation of ice particles and resulting dehydration in the polar winter stratosphere"

_Atmospheric Chemistry and Physics, 2018_

## Referee Comment (RC1) · Anonymous Referee #2 · 18 Jul 2018

The manuscript 'Lagrangian simulation of ice particles and resulting dehydration in the polar winter stratosphere' by Tritscher et al. extends the Chemical Lagrangian Model of the Stratosphere (CLaMS) with a new nucleation and sedimentation scheme for ice PSCs, in addition to existing parameterizations for STS and NAT. The new model version simulates the nucleation, growth, sedimentation, and evaporation of ice PSC particles along individual trajectories and simulates the related dehydration within the ice sedimentation module of CLaMS. The model development can significantly advance the scientific knowledge on polar stratospheric cloud formation. It further enlarges the range of applicability of the CLaMS model to Antarctic polar vortex conditions.

CLaMS results for the Arctic winter 2009/2010 and for the Antarctic winter 2011 are compared to PSC observations from the Cloud-Aerosol Lidar with Orthogonal Polar-

ization (CALIOP), to the Michelson Interferometer for Passive Atmospheric Sounding (MIPAS) and to H2O and HNO3 distributions detected by the Microwave Limb Sounder (MLS). The simulations reasonably reproduce the timing and extent of PSC occurrence inside the vortex in the Arctic winter 2009/2010 and the Antarctic winter 2011, and therefore support the applicability of the stratospheric ice nucleation scheme recently suggested by Engel et al. (2013) to regional and even global scales. This is an important new result that merits publication. Deviations between model results and satellite observations as well as shortcomings of the nucleation schemes (used in CLaMS) are mentioned however this point could be discussed in a more detail to elucidate gaps or open questions in PSC science.

The manuscript represents a substantial contribution to scientific progress on polar stratospheric processes and is well within the scope of Atmospheric Chemistry and Physics. The scientific approach and applied methods are valid and clearly outlined and the results of the manuscript are discussed in a balanced way. The overall presentation is well structured and clear and the language fluent and precise. The quality of figures and text is excellent.

In sum, the paper is an important contribution to polar science and well suited for publication in ACP after the major concerns have been fully addressed by the authors.

Major concerns

(1) In addition to the general agreement of the model and the observations, which is impressive, the deviations of the model and the observations could be addressed in a more comprehensive way in order to highlight and fill gaps in scientific knowledge on PSC microphysics.

This discussion can help to answer the following questions: Are implemented nucleation schemes for NAT and ice sufficient to explain all observations? How could the nucleation schemes be corrected or extended to better cover dehydration and denitrification measured by MLS? Shortcomings in the implemented NAT nucleation pathways

lead to deviations in NAT particle size distributions and therefore to biases in the representation of denitrification in the model. Which processes or modifications could reduce those deviations? Similarly deficits in the ice nucleation schemes lead to smaller coverage of ice PSCs in the model and related reduced dehydration. Which processes could reduce shortcomings of the implemented ice nucleation schemes with respect to coverage with ice PSCs? This discussion could help to increase the scientific relevance of the paper and extend science beyond the state of the art. The results of the discussion should clearly be summarized in the abstract.

(2) A quantification of the deviation of the model results with respect to observations will strengthen the discussion of model agreement / disagreement with observations and will help to quantify uncertainties in the observations. In particular, quantitative deviations in stratospheric water vapor and nitric acid distributions between MLS und ClaMS could be added in a new the panel in Figures 5 and 10. Further a quantitative discussion of the agreement of model results and PSC observations by CALIOP and MIPAS could help to increase the value of the so far more qualitative discussion. Adding contour lines of TNAT and Tice to Figures 3, 7 and 8 could give further insights in data quality from observations and model. Could delta TNAT (or delta Tice) instead of temperatures be shown in Figures 3, 7 and 8 lowermost panel to get independent information on PSC phase or ambient conditions?

(3) The abstract could be rephrased to specifically highlight the results of the study with respect to ice PSCs and dehydration. The first 3 sentences of the abstract are too general and do not cover the content of the manuscript and therefore could be shifted to the introduction. If needed in the abstract, a more specific motivation could be given why Lagrangian simulations of ice PSCs and sedimentation are important. The scope of the abstract is to present the scientific results of this study. Comments with respect to previous work or campaigns (without explanation) could be omitted or shifted in the introduction unless it is urgently needed for a specific result. Quantitative descriptions of model agreement with observations should be given and disagreement could be

discussed in sight of missing processes. Comments (1) and (2) will help to enhance the quality of the abstract, which as rather descriptive at the moment.

Minor comments

P2 l19 Which knowledge gaps exist? Be more specific.

P2 l27 Which gaps, weaknesses and uncertainties exist? Be more specific.

P5 l36 Water equilibrium depends in water partial pressure and ice crystals concentrations/surface areas.

P6 l25 Are the temperature fluctuations used for the NAT nucleation pathway, too?

P10 l8 More information on MLS data and uncertainties could be given.

P11 l14 What causes the MIPAS NAT observations/interference < 15km altitude? Polar cirrus are not measured by MIPAS.

P11 l18 Could you comment on the CALIOP and CLAMS results of total PSC, ice and NAT areas below 13 km altitude ?

P11 l26 Could you comment/quantify the agreement/disagreement between ClaMS and CALIOP?

P11 l26 Could you comment on the deviations in EnhNAT between ClaMS and CALIOP (Figure 3).

P11 l29 What causes the spread in CALIOP data (Figure 3, lowermost row) with respect to ClaMS results?

P12 l33 NAT PSCs do not follow due to data gaps, maybe rephrase.

P12 l35 Could you comment on the disagreement in PSC occurrence below 15 km altitude between CLAMS and CALIOP and MIPAS?

P13 l1 Explanation of results from Figure 7 are missing. Again there are similarities
but also differences in the NAT and ice PSC occurrence in the upper panel and in the scatter in the lowermost panel in Figure 7.

P14 General agreement is reasonable or good. Please now explain in detail deviations between model results and observations in sight of current PSC formation schemes. Which processes are not understood or not covered in the model that help to resolve the deviations?

Figure 5 and 10

Could a new the panel be added in Figures 5 and 10 that quantifies the agreement/deviations between MLS and CLaMS?

Figure 3, 7 and 8

Could the TNAT contour lines be given in Figure 3, 7 and 8? This could help to decide on a bias in NAT occurrence by ClaMS or the observations.

Could the Tice contour lines be given in Figure 3, 7 and 8? This could help to decide on a bias in NAT occurrence by ClaMS or the observations.

Could delta TNAT (or delta Tice) instead of temperatures be shown in Figures 3, 7 and 8 lowermost panel to get independent information on PSC phase and to be independent on altitude/H2O and HNOP3 partial pressures?

N ice manuscript. . ..

---

## Referee Comment (RC2) · Anonymous Referee #1 · 24 Jul 2018

The manuscript by Ines Tritscher and coauthors addresses the problematics of polar stratospheric clouds through Lagrangian simulations performed using an advanced version of CLaMS model featuring a new ice sedimentation module. The simulation covers 2009/2010 Arctic winter, characterized by an unusually strong outbreak of ice PSCs and a typical Antarctic PSC season of 2011. The results of CLaMS simulation are validated using satellite observations of PSCs by CALIOP and MIPAS. The intercomparison is done for vortex-wide temporal evolution of PSC areal coverage as well as for water vapour and nitric acid observed by MLS. A more detailed insight into the performance of CLaMS PSC scheme is provided by comparing individual clouds sampled by CALIOP with model simulations. The simulation is shown to reproduce well the observed extent and timing of the PSC occurrence as well as the related dehy-

dration and denitrification in both hemispheres. The detailed comparisons based on single CALIPSO orbits demonstrate an impressive capacity of CLaMS to match high-resolution PSC profiling by CALIOP. While some discrepancies do remain, the results clearly suggest the applicability of the new CLaMS ice nucleation scheme for studies of PSC formation and their effect on polar vortex composition. Overall, these results represent a valuable contribution to the PSC science. The manuscript is well structured and nicely written, the applied methods are described in a comprehensive way and the presentation quality is excellent. I recommend the article for publication in ACP subject to minor revisions as follows.

p.11, l.13-14. The maximum of PSC occurrence seen by MIPAS at 15 km is explained by the possible contamination of PSC detection by cirrus clouds and/or aerosol remaining in the stratosphere after Sarychev eruption. I think the occurrence of cirrus clouds at this level during winter at high latitudes is too rare to introduce such a strong signal. Post-Sarychev sulfuric aerosol sounds more reasonable however I wonder if this aerosol could also bias the CALIOP PSC detection. I suggest that the authors clarify this point. A more general question on the subject: could the presence of volcanic aerosol in the polar vortex enhance the formation of PSC?

p.13, l.10-11. If I understood correctly this sentence, it suggests that the overestimation of NAT occurrence by CLaMS with respect to CALIOP observations may be caused by denitrification (supposedly underestimated by simulation). However, this statement is at odds with what can be inferred from Fig. 10, where CLaMS produces even stronger denitrification than that derived from MLS observations.

p.13, l.26-28. "The total magnitude of dehydration is slightly smaller in the simulations than in the observations, which agrees with the impression that CLaMS simulations produce less ice than observed." I did not get the same impression. Instead, Fig. 6 rather shows that CLaMS produces at least as much ice PSC as observed by CALIOP or even more.

[Figure]

Fig. 6. There seem to be different upper limits of the color scale in the upper-row plots. Do these plots really have a unique color scale?

[Figure]

---

## Referee Comment (RC3) · Anonymous Referee #3 · 26 Jul 2018

This paper examines an extension to the CLaMS simulations based on a new Lagrangian scheme adding ice PSC particle nucleation, growth, sedimentation, and evaporation. These CLaMS simulations forced with ERA-Interim data are compared with CALIOP, MIPAS and MLS observations for the 2009/2010 Arctic and 2011 Antarctic winters. As identified in the paper the agreement between the CLaMS simulations and the CALIOP, MIPAS, and MLS observations is in general very good over a wide range of temporal and spatial scales and these results suggest this extension of previous work provides improved results and physical insight. This work is clearly of high quality and in my opinion requires only very minor revision before being worthy of being published in ACP. However, I would say that the way that this work has been formed relies less on mathematical rigour than comparison of patterns by eye. Some statistics

identifying the strength of the pattern correlations (that are quite clearly high, but are not quantified) might add some quantification of the quality of the CLaMS output that might aid in future studies to determine whether the model skill has been improved relative to the current study.

In addition, a number of small points that the authors might wish to consider are identified below. However, I should note that many are effectively grammatical.

Minor points:

Abstract: Page 1 Sentence starting on Line 13: This is a slightly confusing sentence, I think you wish to identify that you compare the CLaMS simulations with water vapor data from the MLS observations. But, this sentence is currently in need of revision.

Page 2 Line 6: Replace 'precise and realistic' with 'precisely and realistically'

Page 2 Line 34: Replace 'mid of January' with 'mid January'

Page 4 Sentence starting on Line 9: Replace 'This step enables now the simulation of water redistribution' with 'This step enables the simulation of the water redistribution'

Page 4 Sentence starting on Line 30: Replace 'The idea behind is that particles' with 'The idea behind this is that particles'

First Sentence on Page 6: Maybe mention at this point that the sources of these small scale temperature fluctuations in the atmosphere are often related to gravity waves. I know this is done almost immediately after this point, but it feels like this information needs to be mentioned earlier,.

Sentence starting at Line 29 on Page 26: Maybe should add that at least one paper has done statistical/climatological analysis in both hemispheres. See Alexander, S. P., et al. (2013). "Quantifying the role of orographic gravity waves on polar stratospheric cloud occurrence in the Antarctic and the Arctic." Journal of Geophysical Research-Atmospheres 118(20): 15.

Page 11 Sentence starting on Line 13: is this the signal around 12km? which is poorly represented in the CLaMS ice area? Is this related to the MIPAS PSC classification problem identified later in this section or an unrelated issue?

Page 11 Line 22: Replace 'been in the focus of' with 'been the focus of'

Figure4 text : The text on Page 12 related to Figure 4 mainly focusses on the potential for misclassification of NAT and STS in MIPAS. However, there is also clearly a relatively large discrepancy for ice. Is this also likely related to limitations of the MIPAS retrieval or other factors?

Page 12 Line 23: Replace 'patter' with 'pattern'

Figure 6 test starting on Page 13: Could you explain the origin of the large PSC area at low altitudes (around 12km) seen in CLAMS PSC area panel relative to the CALIOP and MIPAS areas?
* * *

---

## Author Comment (AC1) · 12 Dec 2018

We would like to thank the anonymous reviewer for reading this manuscript and offering suggestions for improvements. In the following, we respond to his/her comments.

Major concern (1) In addition to the general agreement of the model and the observations, which is impressive, the deviations of the model and the observations could be addressed in a more comprehensive way in order to highlight and fill gaps in scientific knowledge on PSC microphysics. This discussion can help to answer the following questions: Are implemented nucleation schemes for NAT and ice sufficient to explain all observations? How could the nucleation

schemes be corrected or extended to better cover dehydration and denitrification measured by MLS? Shortcomings in the implemented NAT nucleation pathways lead to deviations in NAT particle size distributions and therefore to biases in the representation of denitrification in the model. Which processes or modifications could reduce those deviations? Similarly deficits in the ice nucleation schemes lead to smaller coverage of ice PSCs in the model and related reduced dehydration. Which processes could reduce shortcomings of the implemented ice nucleation schemes with respect to coverage with ice PSCs? This discussion could help to increase the scientific relevance of the paper and extend science beyond the state of the art. The results of the discussion should clearly be summarized in the abstract.

We considered your first comment as very important and tried to improve the different parts of the paper (Results, Discussion, and Abstract) accordingly. We have also taken into account your comment on "processes", which are better discussed in the revised version now. However, a detailed study about the importance of individual PSC formation mechanisms is beyond the scope of this paper. This publication is primarily meant as an introduction to the new CLaMS PSC ice module. We already have in mind to come up with a further study analyzing and (hopefully) understanding PSC formation in more detail. However, here is one example with changes to the discussion of the existing manuscript:

"However, the comparisons with CALIOP also shows differences regarding NAT occurrence. Cloud free areas, next to or surrounded by PSCs in the CALIOP data, are often populated with NAT particles in the CLaMS simulations. Looking at the temporal and vortex averaged evolution of  $HNO_3$ , CLaMS shows an uptake of  $HNO_3$  from the gas into the particle phase which is somewhat too large and happens too early in the NH season. The permanent redistribution of  $HNO_3$  in the NH is smaller compared to the observations. Also the Antarctic
model run shows too little denitrification at lower altitudes towards the end of the winter compared to the observations. These findings point to shortcomings in the simulation of NAT particle sizes in combination with number densities namely that NAT particle sedimentation should be more efficient in CLaMS. Further studies should try to find better combinations of NAT number densities and sizes with the potential to denitrify the stratosphere more precisely. Heterogeneous NAT nucleation on foreign nuclei and preexisting ice particles is already implemented and covers most currently discussed routes to form NAT. However, NAT clouds downstream of mountain waves may act as "mother clouds" and individual NAT particles falling out of these clouds in low number densities can grow to large sizes of up to  $10 \,\mu$ m (Fueglistaler et al., 2002). So far, CLaMS comprises the development of high number density NAT clouds on ice surfaces. No attention has been paid to the "mother cloud" theory, which could be a step forward to resolve deviations seen in the NAT simulations."

Major concern (2) A quantification of the deviation of the model results with respect to observations will strengthen the discussion of model agreement / disagreement with observations and will help to quantify uncertainties in the observations. In particular, quantitative deviations in stratospheric water vapor and nitric acid distributions between MLS und CLaMS could be added in a new the panel in Figures 5 and 10. Further a quantitative discussion of the agreement of model results and PSC observations by CALIOP and MIPAS could help to increase the value of the so far more qualitative discussion. Adding contour lines of TNAT and Tice to Figures 3, 7 and 8 could give further insights in data quality from observations and model. Could delta TNAT (or delta Tice) instead of temperatures be shown in Figures 3, 7 and 8 lowermost panel to get independent information on PSC phase or ambient conditions?
As suggested by the reviewer, we added new panels to Figs. 5 and 10 showing the deviations between MLS and CLaMS. For clarity reasons, we removed the panel showing CLaMS results without the MLS averaging kernel. Moreover, we added contour lines of  $T_{\rm NAT}$  and  $T_{\rm frost}$  to the upper row of Figs. 3, 7, and 8 and set the color coded temperatures in the scatter plots in relation to the frost point.

Major concern (3) The abstract could be rephrased to specifically highlight the results of the study with respect to ice PSCs and dehydration. The first 3 sentences of the abstract are too general and do not cover the content of the manuscript and therefore could be shifted to the introduction. If needed in the abstract, a more specific motivation could be given why Lagrangian simulations of ice PSCs and sedimentation are important. The scope of the abstract is to present the scientific results of this study. Comments with respect to previous work or campaigns (without explanation) could be omitted or shifted in the introduction unless it is urgently needed for a specific result. Quantitative descriptions of model agreement with observations should be given and disagreement could be discussed in sight of missing processes. Comments (1) and (2) will help to enhance the quality of the abstract, which is rather descriptive at the moment.

We agree that the abstract was quite descriptive so far and reformulated it completely:

"Polar stratospheric clouds (PSCs) and cold stratospheric aerosols drive heterogeneous chemistry and play a major role in polar ozone depletion. The Chemical Lagrangian Model of the Stratosphere (CLaMS) simulates the nucleation, growth, sedimentation, and evaporation of PSC particles along individual trajectories. Particles consisting of nitric acid trihydrate (NAT), which contain a substantial fraction of the stratospheric nitric acid (HNO3), were the focus of previous modeling work and are known for their potential to denitrify the polar stratosphere.
Here, we carried this idea forward and introduced the formation of ice PSCs and related dehydration into the sedimentation module of CLaMS. Both processes change the simulated chemical composition of the lower stratosphere. Due to the Lagrangian transport scheme, NAT and ice particles move freely in three-dimensional space. Heterogeneous NAT and ice nucleation on foreign nuclei as well as homogeneous ice nucleation and NAT nucleation on preexisting ice particles are now implemented into CLaMS and cover major PSC formation pathways.

We show results from the Arctic winter 2009/2010 and from the Antarctic winter 2011 to demonstrate the performance of the model over two entire PSC seasons. For both hemispheres, we present CLaMS results in comparison to measurements from the Cloud-Aerosol Lidar with Orthogonal Polarization (CALIOP), the Michelson Interferometer for Passive Atmospheric Sounding (MIPAS), and the Microwave Limb Sounder (MLS). Observations and simulations are presented on season-long and vortex-wide scales as well as for single PSC events. The simulations reproduce well both the timing and the extent of PSC occurrence inside the entire vortex. Divided into specific PSC classes, CLaMS results show predominantly good agreement with CALIOP and MIPAS observations, even for specific days and single satellite orbits. CLaMS and CALIOP agree that NAT mixtures are the first type of PSC to be present in both winters. NAT PSC areal coverages over the entire season agree satisfactorily. However, cloud free areas, next to or surrounded by PSCs in the CALIOP data, are often populated with NAT particles in the CLaMS simulations. Looking at the temporal and vortex averaged evolution of HNO3, CLaMS shows an uptake of HNO3 from the gas into the particle phase which is too large and happens too early in the simulation of the Arctic winter. In turn, the permanent redistribution of HNO3 is smaller in the simulations than in the observations. The Antarctic model run shows too little denitrification at lower altitudes towards the end of the winter compared to the observations. The occurrence of synoptic-scale ice PSCs agree satisfactorily
between observations and simulations for both hemispheres and the simulated vertical redistribution of water vapor ( $H_2O$ ) is in very good agreement with MLS observations. In summary, a conclusive agreement between CLaMS simulations and a variety of independent measurements is presented."

**Minor comments**

P2 I19 Which knowledge gaps exist? Be more specific.

We rephrased this general statement to be more specific:

"Due to unknown processes in the formation of solid PSC particles, large differences in the parameterization of PSCs in global models exist."

**P2 127 Which gaps, weaknesses and uncertainties exist? Be more specific.**

We improved this text passage as well:

"Non satisfying agreement between models and observations as well as fundamental differences e.g. in the NAT nucleation exist even in advanced PSC schemes, which further motivated the research presented in this paper."

**P5 126** Water equilibrium depends in water partial pressure and ice crystals concentrations/surface areas.

We agree with this statement and added one more sentence to the manuscript:

"Water equilibrium depends on gas-phase water partial pressure and water vapor pressure of the aerosol particles."

P6 125 Are the temperature fluctuations used for the NAT nucleation pathway, too?
Yes, they are used for both nucleation pathways, NAT and ice. We clarified this at the end of Section 2.2.

P10 18 More information on MLS data and uncertainties could be given.

We added information about the A-Train satellite constellation and MLS uncertainties.

P11 I14 What causes the MIPAS NAT observations/interference

sified as NAT (Spang et al., 2018). In CLaMS, we do not simulate clouds other than PSCs. The origin of the larger PSC area at low altitudes seen in the CLaMS PSC area panel can be explained by the altitude independent fixed detection threshold of  $3.3 \,\mu$  m2 cm-3 for STS droplets. At altitudes around 12 km, the stratospheric aerosol layer becomes visible as well. To reduce the large "PSC area" in CLaMS at low altitudes, we introduced a temperature threshold to this plot. Only data points with temperatures less than 200 K are considered. This temperature threshold reduces the maximum values of PSC areal coverage slightly."

P11 118 Could you comment on the CALIOP and CLAMS results of total PSC, ice and NAT areas below 13 km altitude?

Please see my comment to P11 I14 above.

P11 126 Could you comment/quantify the agreement/disagreement between CLaMS and CALIOP?

We understand that our discussion about the shown figure is on the short side. Therefore, we added more details to our explanation of Fig. 3 in the manuscript.

P11 126 Could you comment on the deviations in EnhNAT between CLaMS and CALIOP (Figure 3).

We added the following paragraph:

"Enhanced NAT mixtures represent PSCs heterogeneously nucleated in wave ice PSCs. The CALIOP criteria defining enhanced NAT mixtures are conservative and therefore, the enhanced NAT mixtures subclass is not all-inclusive (Pitts et al., 2018). On this particular day, we expect no NAT PSCs downstream of wave ice clouds. Whereas this area is not populated in the CLaMS data, single scattered measurement points from CALIOP fall into this class, likely due to
P11 129 What causes the spread in CALIOP data (Figure 3, lowermost row) with respect to CLaMS results?

We added the following information:

"The spread in the CALIOP data is caused by measurement noise. Although measurement noise is mimicked and added to the modeled data, the spread in the modeled data is slightly less than for the observed values. Those data points are still more confined and do not fill the whole space of the diagram."

P12 I33 NAT PSCs do not follow due to data gaps, maybe rephrase.

We rephrased this sentence.

P12 I35 Could you comment on the disagreement in PSC occurrence below 15 km altitude between CLAMS and CALIOP and MIPAS?

Please see my comment to P11 I14 above.

**P13 I1** Explanation of results from Figure 7 are missing. Again there are similarities but also differences in the NAT and ice PSC occurrence in the upper panel and in the scatter in the lowermost panel in Figure 7.

We extended the description of Fig. 7 in the manuscript.

P14 General agreement is reasonable or good. Please now explain in detail deviations between model results and observations in sight of current PSC formation Interactive comment

schemes. Which processes are not understood or not covered in the model that help to resolve the deviations?

Please see my comment to your Major concern (2).

Figure 5 and 10 Could a new panel be added in Figures 5 and 10 that quantifies the agreement/deviations between MLS and CLaMS?

Done. Please see Fig. 1 below.

Figure 3, 7 and 8 Could the TNAT contour lines be given in Figure 3, 7 and 8? This could help to decide on a bias in NAT occurrence by CLaMS or the observations. Could the Tice contour lines be given in Figure 3, 7 and 8? This could help to decide on a bias in NAT occurrence by CLaMS or the observations. Could delta TNAT (or delta Tice) instead of temperatures be shown in Figures 3. 7 and 8 lowermost panel to get independent information on PSC phase and to be independent on altitude/ $H_2O$  and  $HNO_3$  partial pressures?

Please see Fig. 2 below.

We added contour lines for  $T_{NAT}$  and  $T_{frost}$  and adopted also the temperatures shown in the lowermost panels of the corresponding figures. However, we would like to note that those temperatures depend on temperatures, pressure levels, and vapor concentrations from CLaMS and could therefore easily be wrong by some few Kelvin as well.

**References**

Fueglistaler, S., Luo, B. P., Voigt, C., Carslaw, K. S., and Peter, T.: NAT-rock formation by mother clouds: a microphysical model study, Atmos. Chem. Phys., 2, 93-98, https://doi.org/

ACPD
**Printer-friendly version**

10.5194/acp-2-93-2002, 2002.

Pitts, M. C., Poole, L. R., and Gonzalez, R.: Polar stratospheric cloud climatology based on CALIPSO spaceborne lidar measurements from 2006 to 2017, Atmos. Chem. Phys., 18, 10881–10913, https://doi.org/10.5194/acp-18-10881-2018, 2018.

Spang, R., Hoffmann, L., Müller, R., Grooß, J.-U., Tritscher, I., Höpfner, M., Pitts, M., Orr, A., and Riese, M.: A climatology of polar stratospheric cloud composition between 2002 and 2012 based on MIPAS/Envisat observations, Atmos. Chem. Phys., 18, 5089–5113, https://doi.org/10.5194/acp-18-5089-2018, 2018.
**ACPD**

---

## Author Comment (AC2) · 12 Dec 2018

We would like to thank the anonymous reviewer for reading this manuscript and offering suggestions for improvements. In the following, we respond to his/her comments.

**Minor comments**

**p.11, l.13-14.** *The maximum of PSC occurrence seen by MIPAS at 15 km is explained by the possible contamination of PSC detection by cirrus clouds and/or aerosol remaining in the stratosphere after Sarychev eruption. I think the occurrence of cirrus clouds at this level during winter at high latitudes is too rare to introduce*

[Figure]

*such a strong signal. Post-Sarychev sulfuric aerosol sounds more reasonable however I wonder if this aerosol could also bias the CALIOP PSC detection. I suggest that the authors clarify this point. A more general question on the subject: could the presence of volcanic aerosol in the polar vortex enhance the formation of PSC?*

To reply to this comment, mentioned in all three reviews, we further improved Figs. 2 and 6 of the manuscript. The ACPD version shows solely PSC clouds detected by CALIOP and simulated by CLaMS. However, the MIPAS data include cirrus clouds as well, even though they are often misclassified as NAT. Therefore, we now include cirrus data from the CALIOP data set. By doing this, it becomes evident that CALIOP observes cirrus clouds throughout the entire 2009/2010 season at altitudes below 15 km. CALIOP also observes some NAT mixtures at lower altitudes, but these are likely cirrus that have been misclassified. In reference to the comment on volcanic aerosol, MIPAS is highly sensitive to volcanic aerosol whereas CALIOP will consider volcanic aerosol as part of the "background". If the aerosol is widespread, it will not be included in the CALIOP PSC product since it just identifies outliers. Only if it is a localized plume, then it would be identified as PSC. In CLaMS, we do not simulate clouds other than PSCs. The origin of the large PSC area at low altitudes seen in the CLaMS PSC area panel can be explained by the altitude independent fixed detection threshold of $3.3\,\mu\mathrm{m}^2\,\mathrm{cm}^{-3}$ for STS droplets. At altitude levels around 12 km, the Junge layer becomes visible as well. To reduce the large "PSC area" in CLaMS at low altitudes, we introduced now a temperature threshold to this plot. Only data points with temperatures less than 200 K are considered. This temperature threshold reduces the maximum values slightly.

We explained this now in more detail in the paper as well.

To the last more general question: In our understanding, the presence of volcanic aerosol in the polar vortex would enhance PSC formation at least in

the northern hemisphere. It would increase the number of heterogeneous nuclei and therefore increase the probability of NAT and/or ice nucleation. Of course, temperatures and the availability of water and nitric acid plays a role, too. In the southern hemisphere, where temperature are well below the frost point anyway, the additional presence of volcanic aerosol might not change the total number of PSCs. It might lead to an earlier occurrence of PSCs in the season. However, this question could be the focus of further, detailed research.

**p.13, l.10-11.** *If I understood correctly this sentence, it suggests that the overestimation of NAT occurrence by CLaMS with respect to CALIOP observations may be caused by denitrification (supposedly underestimated by simulation). However, this statement is at odds with what can be inferred from Fig. 10, where CLaMS produces even stronger denitrification than that derived from MLS observations.*

We realized that our explanation was too short. Therefore, we tried to make this point more clear.

"A comparison between MLS and CLaMS HNO$_3$ mixing ratios is acceptable but reveals differences. CLaMS HNO$_3$ gas phase mixing ratios around 500 K potential temperature are lower than the observations for the whole season. However, from August on, a layer of high HNO$_3$ values below 500 K points to the possibility that the redistribution of HNO$_3$ is not efficient enough in the simulation and needs to extend down to lower altitudes. This might explain the simulation of NAT particles in areas which are almost cloud free in the observations as seen in Fig. 7. Even though CLaMS gas phase mixing ratios of HNO$_3$ might be even lower than observed at that time, HNO$_3$ in the model could still be present in the particle phase and could not be redistributed correctly to lower altitudes."

**p.13, l.26-28.** *"The total magnitude of dehydration is slightly smaller in the simulations*

*than in the observations, which agrees with the impression that CLaMS simulations produce less ice than observed." I did not get the same impression. Instead, Fig. 6 rather shows that CLaMS produces at least as much ice PSC as observed by CALIOP or even more.*

After producing a new panel for Fig. 10, showing the quantitative deviations in stratospheric water vapor between MLS and CLaMS, we now agree with this comment. We therefore changed the text in the manuscript accordingly.

"Overall, over the entire season, CLaMS simulations somewhat underestimate ice occurrences on several occasions (e.g. Fig. 8, July and August). However, Fig. 6 gives the impression that the areal coverage of ice PSCs is at least as large as in the observations. [...] The temporal evolution of gas-phase water vapor and nitric acid as measured by MLS and simulated by CLaMS is presented in Fig. 10. [...] The difference between measurements and simulations are quantified in the right panels (Fig. 10). The minimum values of $H_2O$ match very well. The layer of rehydrated air around 350 K potential temperature is slightly less than in the observations meaning that $H_2O$ mixing ratios are smaller in the simulation than in the observations."

**Fig. 6.** *There seem to be different upper limits of the color scale in the upper-row plots. Do these plots really have a unique color scale?*

They do have the same color scale. Only the maximum value differs and is written as upper limit on top of the color scale. We repeated the hint "Please note that the color code is always identical except the maximum value of the top color bin." given at Fig. 2 also for Fig. 6.

---

## Author Comment (AC3) · 12 Dec 2018

We would like to thank the anonymous reviewer for reading this manuscript and offering suggestions for improvements. In the following, we respond to his/her comments.

**Minor comments**

**Abstract: Page 1 Sentence starting on Line 13:** *This is a slightly confusing sentence, I think you wish to identify that you compare the CLaMS simulations with water vapor data from the MLS observations. But, this sentence is currently in need of revision.*

We revised the sentence and parts of the Abstract as a consequence of the major concern (3) from Anonymous Referee #2.

**Page 2 Line 6:** *Replace "precise and realistic" with "precisely and realistically"*

Done.

**Page 2 Line 34:** *Replace "mid of January" with "mid January"*

Done.

**Page 4: Sentence starting on Line 9:** *Replace "This step enables now the simulation of water redistribution" with "This step enables the simulation of the water redistribution"*

Done.

**Page 4: Sentence starting on Line 30:** *Replace "The idea behind is that particles" with "The idea behind this is that particles"*

Done.

**First Sentence on Page 6:** *Maybe mention at this point that the sources of these small scale temperature fluctuations in the atmosphere are often related to gravity waves. I know this is done almost immediately after this point, but it feels like this information needs to be mentioned earlier.*

Done.

**Sentence starting at Line 29 on Page 26:** *Maybe should add that at least one paper has done statistical/climatological analysis in both hemispheres. See Alexander, S. P., et al. (2013). "Quantifying the role of orographic gravity waves on polar stratospheric cloud occurrence in the Antarctic and the Arctic." Journal of Geophysical Research - Atmospheres 118(20): 15.*

We added this citation.

**Page 11 Sentence starting on Line 13:** *Is this the signal around 12 km? which is poorly represented in the CLaMS ice area? Is this related to the MIPAS PSC classification problem identified later in this section or an unrelated issue?*

To reply to this comment, mentioned in all three reviews, we further improved Figs. 2 and 6 of the manuscript. The ACPD version shows solely PSC clouds detected by CALIOP and simulated by CLaMS. However, the MIPAS data include cirrus clouds as well, even though they are often misclassified as NAT. Therefore, we now include cirrus data from the CALIOP data set. By doing this, it becomes evident that CALIOP observes cirrus clouds throughout the entire 2009/2010 season at altitudes below 15 km. CALIOP also observes some NAT mixtures at lower altitudes, but these are likely cirrus that have been misclassified. In reference to the comment on volcanic aerosol, MIPAS is highly sensitive to volcanic aerosol whereas CALIOP will consider volcanic aerosol as part of the "background". If the aerosol is widespread, it will not be included in the CALIOP PSC product since it just identifies outliers. Only if it is a localized plume, then it would be identified as PSC. In CLaMS, we do not simulate clouds other than PSCs.

We explained this now in more detail in the paper as well.

**Page 11 Line 22:** *Replace "been in the focus of" with "been the focus of"*

Done.

**Figure 4 text:** *The text on Page 12 related to Figure 4 mainly focuses on the potential for misclassification of NAT and STS in MIPAS. However, there is also clearly a relatively large discrepancy for ice. Is this also likely related to limitations of the MIPAS retrieval or other factors?*

Taking into account that the measured volume of MIPAS and the sampled volume of the model does not perfectly match and that small temperature deviations in the model compared to reality matter a lot if one looks at ice formation, we still think that the comparison of ice PSC occurrence between MIPAS and CLaMS agrees well. The spatial pattern of ice occurrence has been reproduced by CLaMS as well as the majority of single ice observations. However, we changed the text in the manuscript slightly. "Even though ice formation is highly temperature depended, the spatial pattern of ice PSC occurrence between MIPAS and CLaMS agrees well (Fig. 4)."

**Page 12 Line 23:** *Replace "patter" with "pattern"*

Done.

**Figure 6 text starting on Page 13:** *Could you explain the origin of the large PSC area at low altitudes (around 12 km) seen in CLAMS PSC area panel relative to the CALIOP and MIPAS areas?*

This point has been mentioned in all three reviews, too. The origin of the large PSC area at low altitudes seen in the CLaMS PSC area panel can be explained by the altitude independent fixed detection threshold of $3.3 \, \mu m^2 \, cm^{-3}$ for STS droplets. At altitude levels around 12 km, the Junge layer becomes visible as

well. To reduce the large "PSC area" in CLaMS at low altitudes, we introduced now a temperature threshold to this plot. Only data points with temperatures less than 200 K are considered. This temperature threshold reduces the maximum values slightly.
* * *

---

## Author Response (AR1)

**Author's response**

We thank the handling Co-Editor Farahnaz Khosrawi for her time and effort, which she spent on our manuscript. We also thank once more the three anonymous reviewers for reading the manuscript and offering suggestions for improvements. We provided a detailed point-by-point response to all referee comments in the interactive discussion of the paper and do not repeat our answers here. Instead, we prepared a marked-up manuscript version showing the changes made. Please note, that we performed also changes to Figs. 2, 3, 5, 6, 7, 8, and 10.

[revised manuscript text omitted]